# Variance Reduction via Accelerated Dual Averaging for Finite-Sum Optimization

**Chaobing Song**[†*]        **Yong Jinag**[†]        **Yi Ma**[‡]

[†]Tsinghua-Berkeley Shenzhen Institute, Tsinghua University
songcb16@mails.tsinghua.edu.cn,    jiangy@sz.tsinghua.edu.cn
[‡]Department of EECS, University of California, Berkeley
yima@eecs.berkeley.edu

## Abstract

In this paper, we introduce a simplified and unified method for finite-sum convex optimization, named *Variance Reduction via Accelerated Dual Averaging (VRADA)*. In both general convex and strongly convex settings, VRADA can attain an $O\left(\frac{1}{n}\right)$-accurate solution in $O(n \log \log n)$ number of stochastic gradient evaluations which improves the best known result $O(n \log n)$, where $n$ is the number of samples. Meanwhile, VRADA matches the lower bound of the general convex setting up to a $\log \log n$ factor and matches the lower bounds in both regimes $n \leq \Theta(\kappa)$ and $n \gg \kappa$ of the strongly convex setting, where $\kappa$ denotes the condition number. Besides improving the best known results and matching all the above lower bounds simultaneously, VRADA has more unified and simplified algorithmic implementation and convergence analysis for both the general convex and strongly convex settings. The underlying novel approaches such as the novel initialization strategy in VRADA may be of independent interest. Through experiments on real datasets, we show the good performance of VRADA over existing methods for large-scale machine learning problems.

## 1   Introduction

In this paper, we study the following composite convex optimization problem:

$$\min_{\boldsymbol{x} \in \mathbb{R}^d} f(\boldsymbol{x}) := g(\boldsymbol{x}) + l(\boldsymbol{x}) := \frac{1}{n} \sum_{i=1}^{n} g_i(\boldsymbol{x}) + l(\boldsymbol{x}), \tag{1}$$

where $g(\boldsymbol{x}) := \frac{1}{n} \sum_{i=1}^{n} g_i(\boldsymbol{x})$ with $g_i(\boldsymbol{x})$ being convex and smooth, and $l(\boldsymbol{x})$ is convex, probably nonsmooth but admitting an efficient proximal operator. In this paper, we mainly assume that each $g_i(\boldsymbol{x})$ is $L$-smooth ($L > 0$) and $l(\boldsymbol{x})$ is $\sigma$-strongly convex ($\sigma \geq 0$). If $\sigma = 0$, then the problem is *general convex*. If $\sigma > 0$, then the problem is *strongly convex* and we define the corresponding condition number $\kappa := L/\sigma$. Instances of problem (1) appear widely in statistical learning, operations research, and signal processing. For instance, in machine learning, if $\forall i \in [n], g_i(\boldsymbol{x}) := h_i(\langle \boldsymbol{a}_i, \boldsymbol{x} \rangle)$, where $h_i : \mathbb{R} \to \mathbb{R}$ is a convex loss function and $\boldsymbol{a}_i \in \mathbb{R}^d$ is the data vector, then the problem (1) is also called regularized *empirical risk minimization* (ERM). Important instances of ERM include ridge regression, Lasso, logistic regression, and support vector machine.

In the large-scale setting where $n$ is large, first-order methods become the natural choice for solving (1) due to its better scalability. However, when $n$ is very large, even accessing the full gradient $\nabla g(\boldsymbol{x})$ becomes prohibitively expensive. To alleviate this difficulty, a common approach is to use an unbiased stochastic gradient $\nabla g_i(\boldsymbol{x})$ ($i$ is randomly chosen from $[n] := \{1, 2, \ldots, n\}$) to replace

---

[*]This work was conducted during Chaobing Song's visit to Professor Yi Ma's group at UC Berkeley.

Table 1: Complexity results for solving the problem (1) with accuracy $\epsilon \geq \Theta(L/n)$.

| Algorithm | General/Strongly Convex |
|---|---|
| SVRG$^{++}$ [4] | $O\big(n \log \frac{1}{\epsilon}\big)$ |
| Varag [19] | $O\big(n \log \frac{1}{\epsilon}\big)$ |
| VRADA (**This Paper**) | $O\big(n \log \log \frac{1}{\epsilon}\big)$ |
| Lower bound [37] | $\Omega(n)$ |

the full gradient $\nabla g(\boldsymbol{x})$ in each iteration, *a.k.a.*, *stochastic gradient descent (SGD)*. In the stochastic setting, the goal to solve (1) becomes to find an expected $\epsilon$-accurate solution $\boldsymbol{x} \in \mathbb{R}^d$ satisfying $\mathbb{E}[f(\boldsymbol{x})] - f(\boldsymbol{x}^*) \leq \epsilon$, where $\boldsymbol{x}^*$ is an exact minimizer of (1). Typically, the iteration complexity result of such an algorithm is evaluated by the number of evaluating stochastic gradients $\nabla g_i(\boldsymbol{x})$ needed to achieve the $\epsilon$-accurate solution.

By only accessing stochastic gradients, SGD has a low per-iteration cost. However, SGD has a very high iteration complexity due to the constant variance $\|\nabla g_i(\boldsymbol{x}) - \nabla g(\boldsymbol{x})\|$. To reduce the iteration complexity of SGD while still maintaining its low per-iteration cost, a remarkable progress in the past decade is to exploit the finite-sum structure of $g$ in (1) to reduce the variance of stochastic gradients. In such *variance reduction* methods, instead of directly using $\nabla g_i(\boldsymbol{x})$, we compute a full gradient $\nabla g(\tilde{\boldsymbol{x}})$ of an anchor point $\tilde{\boldsymbol{x}}$ beforehand. Then we use the following variance reduced gradient

$$\tilde{\nabla} g_i(\boldsymbol{x}) := \nabla g_i(\boldsymbol{x}) - \nabla g_i(\tilde{\boldsymbol{x}}) + \nabla g(\tilde{\boldsymbol{x}}) \tag{2}$$

as a proxy for the full gradient $\nabla g(\boldsymbol{x})$ during each iteration. As a result, the amortized per-iteration cost is still the same as SGD. However, the variance reduced gradient (2) is unbiased and can reduce the variance from $\|\nabla g_i(\boldsymbol{x}) - \nabla g(\boldsymbol{x})\|$ to $\|\nabla g_i(\boldsymbol{x}) - \nabla g_i(\tilde{\boldsymbol{x}})\|$. As the variance $\|\nabla g_i(\boldsymbol{x}) - \nabla g_i(\tilde{\boldsymbol{x}})\|$ can vanish asymptotically, the iteration complexity of SGD can be substantially reduced.

To this end, SAG [32] is historically the first direct[2] variance reduction method to solve (1) while it uses a biased estimation of the full gradient. SVRG [16] directly solves (1) and explicitly uses the unbiased estimation (2) to reduce variance. Then SAGA [10] provides an alternative of (2) to avoid precomputing the gradient of an anchor point but with the price of an increased memory cost. Based on [35], a Catalyst approach [23] has been proposed to combine Nesterov's acceleration into variance reduction methods in a black box manner. [2] has proposed the first direct approach, named Katyusha (*a.k.a.* accelerated SVRG), to combine variance reduction and a kind of Nesterov's acceleration scheme in a principled manner. [37] has given a tight lower complexity bound for finite-sum stochastic optimization and shown the tightness of Katyusha (with black-box reduction [3]) up to a logarithmic factor. MiG [40] follows and simplifies Katyusha by two-point coupling to produce acceleration. Varag [19] improves Katyusha further by considering a unified approach for both the general convex and strongly convex settings. Finally, [13] has proved improved convergence results for a variant of SVRG when $n \gg \kappa$, which is better than the best known results of accelerated ones such as Katyusha.

## 1.1 Related Results and Our Contributions

In Table 1, for clarity, we list the state of the art results (as well as results of this paper and lower bounds [37]) for attaining an accuracy $\epsilon \geq \Theta\big(\frac{L}{n}\big)$. In Table 2, we give the complexity results of representative *direct* variance reduction methods for both the general convex and strongly convex settings (as well as results of this paper and lower bounds). The literature on variance reduction is too rich to list them all here.[3] In Table 2, we mainly list the algorithms with improved convergence results for at least one setting.

To understand where we stand with these complexity results, firstly we are particularly interested in attaining a solution with a proper accuracy such as $\epsilon = \Theta\big(\frac{L}{n}\big)$.[4] To attain this accuracy, as shown

Table 2: Complexity results for solving the problem (1). ("—" means the corresponding result does not exist or is unknown.)

| Algorithm | General Convex | Strongly Convex $n \leq \Theta(\kappa)$ | Strongly Convex $n \gg \kappa$ |
|---|---|---|---|
| SAG [32] | — | $O\left(n\kappa \log \frac{1}{\epsilon}\right)$ | $O\left(n \log \frac{1}{\epsilon}\right)$ |
| SVRG [16, 38, 13] | — | $O\left(\kappa \log \frac{1}{\epsilon}\right)$ | $O\left(n + \frac{n}{\log(n/\kappa)} \log \frac{1}{\epsilon}\right)$ |
| SAGA [10] | $O\left(\frac{n+L}{\epsilon}\right)$ | $O(\kappa \log \frac{1}{\epsilon})$ | $O\left(n \log \frac{1}{\epsilon}\right)$ |
| SVRG$^{++}$ [4] | $O\left(n \log \frac{1}{\epsilon} + \frac{L}{\epsilon}\right)$ | — | — |
| Katyusha$^{sc}$ [2] | — | $O\left(\sqrt{n\kappa} \log \frac{1}{\epsilon}\right)$ | $O\left(n \log \frac{1}{\epsilon}\right)$ |
| Katyusha$^{nsc}$ [2] | $O\left(\frac{n+\sqrt{nL}}{\sqrt{\epsilon}}\right)$ | — | — |
| Varag [19][1] | $O\left(n \log n + \frac{\sqrt{nL}}{\sqrt{\epsilon}}\right)$ | $O\left(n \log n + \sqrt{n\kappa} \log \frac{1}{\kappa\epsilon}\right)$ [2] | $O(n \log \frac{1}{\epsilon})$ |
| VRADA[1] (**This Paper**) | $O\left(n \log\log n + \frac{\sqrt{nL}}{\sqrt{\epsilon}}\right)$ | $O\left(n \log\log n + \sqrt{n\kappa} \log \frac{1}{n\epsilon}\right)$ [3] | $O\left(n + \frac{n}{\log(n/\kappa)} \log \frac{1}{\epsilon}\right)$ [4] |
| Lower bound [37] | $\Omega\left(n + \frac{\sqrt{nL}}{\sqrt{\epsilon}}\right)$ | $\Omega\left(n + \sqrt{n\kappa} \log\left(\sqrt{\frac{n}{\kappa}}\frac{1}{\epsilon}\right)\right)$ | $O\left(n + \frac{n}{\log(n/\kappa)} \log \frac{1}{\epsilon}\right)$ [5] |

[1] For both Varag and VRADA, the complexity results are given for accuracy $\epsilon < \Theta(L/n)$. (For $\epsilon \geq \Theta(L/n)$, see Table 1.)

[2] For more precise bounds of Varag, see [19].

[3] For this setting, a slightly worse but simpler bound is $O(n + \sqrt{n\kappa} \log \frac{1}{\epsilon})$.

[4] For this setting, the $O\left(n \log\log n + \frac{n}{\log(n/\kappa)} \log \frac{1}{n\epsilon}\right)$ is also valid.

[5] This lower bound is only valid for the class of "oblivious p-CLI" algorithms [5, 13].

in Table 1, for both general/strongly convex settings, the non-accelerated SVRG$^{++}$ and accelerated Varag need $O(n \log n)$[5] number of iterations whereas the lower bound [37] implies that we may only need $\Omega(n)$ iterations. Before this work, it is not known whether the logarithmic factor gap can be further reduced or not.

As shown in Table 2, in the *general convex setting*, the best known rate is $O\left(n \log n + \frac{\sqrt{nL}}{\sqrt{\epsilon}}\right)$ [6]. In the *strongly convex setting* with $n \leq \Theta(\kappa)$, as shown in Table 2, for small $\epsilon$, both the complexity results of Katyusha$^{sc}$ and Varag can match the lower bound $\Omega\left(n + \sqrt{n\kappa} \log\left(\sqrt{\frac{n}{\kappa}}\frac{1}{\epsilon}\right)\right)$ for any randomized algorithms with "gradient and proximal operator" oracle [37]. When $n \gg \kappa$ (which is common in the statistical learning context such as $\kappa = O(\sqrt{n})$ [8]), a widely known complexity result is $O(n \log \frac{1}{\epsilon})$, attained by both non-accelerated and accelerated methods. However, [13] showed that a variant of SVRG with different parameter settings has a better iteration complexity $O\left(n + \frac{n}{\log(n/\kappa)} \log \frac{1}{\epsilon}\right)$ than $O(n \log \frac{1}{\epsilon})$. The bound $O\left(n + \frac{n}{\log(n/\kappa)} \log \frac{1}{\epsilon}\right)$ is proved to be optimal for the class of so called "oblivious p-CLI algorithms" [5, 13], despite the fact that the bound involves large constants. So the situation seems to be: before this work, there exists no single algorithm that can match the lower bounds of the three settings simultaneously in Table 2. Meanwhile, for accelerated methods, [2, 40] can not unify the general convex/strongly convex settings, while [19] unifies both settings with very complicated parameter settings and thus is not very practical.

**Efficiency.** As shown in Table 1, to attain a solution with $\epsilon \geq \Theta\left(\frac{L}{n}\right)$, the proposed *Variance Reduction via Accelerated Dual Averaging (VRADA)* algorithm only needs $O\left(n \log\log \frac{1}{\epsilon}\right)$ number of iterations, while the best known result is $O\left(n \log \frac{1}{\epsilon}\right)$.

In the general convex setting, as shown in Table 2, to attain an accuracy $\epsilon < \Theta\left(\frac{L}{n}\right)$, our VRADA method achieves the iteration complexity

$$O\left(n \log\log n + \frac{\sqrt{nL}}{\sqrt{\epsilon}}\right),\tag{3}$$

which matches the lower bound up to a $\log\log$ factor, while the best known result before is $O\left(n \log n + \frac{\sqrt{nL}}{\sqrt{\epsilon}}\right)$. As practically speaking, the $\log\log$ factor can be treated as a small constant: for instance

when $n \le 2^{64}$, we have $n \log \log n \le 6n$. Thus, for general convex problems, VRADA can attain an $\Theta\left(\frac{L}{n}\right)$-accurate solution with essentially $O(n)$ iterations, practically matches the lower bound!

In the strongly convex setting with $n \le \Theta(\kappa)$ and $\epsilon < \Theta\left(\frac{L}{n}\right)$, the bound of VRADA becomes

$$O\Big( n \log \log n + \sqrt{n\kappa} \log \frac{1}{n\epsilon} \Big), \tag{4}$$

which is slightly better than the simpler bound $O\big(n + \sqrt{n\kappa} \log \frac{1}{\epsilon}\big)$ as $n \log \log n \le \sqrt{n\kappa} \log n$. Meanwhile, it also matches the corresponding lower bound for small $\epsilon > 0$.

In the strongly convex setting with $n \gg \kappa$ and $\epsilon < \Theta\left(\frac{L}{n}\right)$, the rate of VRADA becomes

$$O\Big( n + \frac{n}{\log(n/\kappa)} \log \frac{1}{n\epsilon} \Big), \quad \text{or} \ \ O\Big( n \log \log n + \frac{n}{\log(n/\kappa)} \log \frac{1}{n\epsilon} \Big), \tag{5}$$

which matches the lower bound for the class of "oblivious p-CLI algorithms" [13]. Compared with the best-known result [13] for the non-accelerated SVRG, VRADA involves very intuitive parameter settings and thus has small constants in the bound (5). So we can say, VRADA matches the lower bounds of the three settings simultaneously for the first time.

**Simplicity.** VRADA follows the framework of MiG [40], thus it only needs two-point coupling in the inner iteration rather than three-point coupling in Katyusha and Varag. Furthermore, similar to MiG, VRADA only needs to keep track of one variable vector in the inner loop, which gives it a better edge in sparse and asynchronous settings [40] than Katyusha and Varag. In the general convex setting, VRADA is also a direct solver without any extra effort to attain the improved complexity result (3). In the strongly convex setting, VRADA attains the optimal results (4) and (5) by using a natural uniform average, fixed and intuitive inner number of iterations and consistent parameter settings for all the epochs, while Katyusha$^{sc}$ and MiG$^{sc}$ use a weighted average, and Varag uses different parameter settings for the first $\Theta(\log n)$ epochs and the other epochs respectively.

**Unification.** VRADA uses the same parameter setting for both the general convex and strongly convex settings. The only difference is that in VRADA, we set the parameter $\sigma = 0$ in the general convex setting, while we set $\sigma > 0$ in the strongly convex setting. Meanwhile, based on a "generalized estimation sequence", we conduct a unified convergence analysis for both settings. The only difference is that the values of two predefined sequences of positive numbers are different. Correspondingly, Katyusha$^{sc}$ and Katyusha$^{nsc}$ (as well as MiG$^{sc}$ and MiG$^{nsc}$) use different parameter settings and independent convergence analysis for both the general convex and strongly convex settings. Varag provides a unified approach for both settings. However to adapt to both settings, the parameter settings of Varag are very complicated and cannot even be stated in the algorithm description.

## 1.2 Our Approach

**Separation of Nesterov's Acceleration and Variance Reduction.**[7] To combine Nesterov's acceleration and variance reduction, [2] has introduced negative momentum to make Nesterov's acceleration and variance reduction *coexist in the inner loop*. Since [2], all the follow-up methods [40, 19] consider similar ideas. However, as results, the resulting convergence analysis becomes complicated and a weighted averaging in the outer iteration is necessary for the strongly convex setting. In this paper, we consider a very different approach instead: let Nesterov's acceleration occur in the outer iteration and variance reduction occur in the inner iteration, separately. This approach makes the convergence analysis significantly simplified and only uniform average needed for the strongly convex setting.

**Novel Initialization to Cancel Randomized Error.** In SVRG-style variance reduction methods, we need to determine the number of inner iterations. The most intuitive implementation of variance reduction methods is using a fixed number of inner iterations (*e.g.,* $\Theta(n)$). However, such a natural choice makes Katyusha (as well as MiG [40]) incur accumulated randomized errors, which makes it converge at a suboptimal rate $O\left(\frac{n+\sqrt{nL}}{\sqrt{\epsilon}}\right)$ in the general convex setting. To alleviate this situation, one may consider an indirect black-box reduction approach [3] or an approach of half SVRG$^{++}$ and half SVRG [19] (*i.e.,* exponentially increasing until a given threshold) to reduce the complexity result

to $O\big(n\log\frac{1}{\epsilon} + \frac{\sqrt{nL}}{\sqrt{\epsilon}}\big)$, which makes both implementation and analysis complicated. In this paper, we consider a rather simplified and effective approach: we only do a (full) gradient descent step and a particular initialization of estimation sequence *before* entering into the main loop. With this approach, we can simply use a fixed number of inner iterations and reduce the complexity result to $O\big(n\log\log\frac{1}{\epsilon} + \frac{\sqrt{nL}}{\sqrt{\epsilon}}\big)$.

**Dual Averaging to Accumulate Strong Convexity.** The most common implementations of accelerated variance reduction methods are variants of (proximal) accelerated mirror descent (AMD) [2, 40, 19]. In the strongly convex setting, AMD-based methods only exploit the strong convexity in the current iteration but still maintains the optimal dependence on $\epsilon$. However, the dependence on $n$ for these methods is not optimal when $n \gg \kappa$. In this paper, we consider an accelerated dual averaging (ADA) approach [26]. AMD and ADA are often viewed as two different kinds of generalizations for Nesterov's accelerated gradient descent, while ADA can exploit the strong convexity along the whole optimization trajectory. As a result, when $n \gg \kappa$, the resulting VRADA algorithm can improve the best known result $O\big(n\log\frac{1}{\epsilon}\big)$ of AMD-based methods by a log factor to $O\big(n + \frac{n}{\log(n/\kappa)}\log\frac{1}{\epsilon}\big)$.

### 1.3 Other Related Works

**Regarding the Lower Bound under Sampling with Replacement.** When the problem (1) is $\sigma$-strongly convex and $L$-smooth with $\kappa = L/\sigma$, [20] has provided a stronger lower bound than [37] such that to find an $\epsilon$-accurate solution $\boldsymbol{x}$ such that $\mathbb{E}[\|\boldsymbol{x} - \boldsymbol{x}^*\|^2] \leq \epsilon$, any randomized incremental gradient methods need at least

$$\Omega\Big(\big(n + \sqrt{n\kappa}\big)\log\frac{1}{\epsilon}\Big) \tag{6}$$

number of iterations when the dimension $d$ is sufficiently large. Our second upper bound in (5) is measured by $\mathbb{E}[f(\boldsymbol{x})] - f(\boldsymbol{x}^*)$. By the strong convexity, when $n \gg \kappa$ and $\epsilon \leq \Theta(1/n)$, if we convert to the Euclidean distance $\mathbb{E}[\|\boldsymbol{x} - \boldsymbol{x}^*\|^2]$, then our rate will be $O\Big(n\log\log n + \frac{n\log(\kappa/(n\epsilon))}{\log(n/\kappa)}\Big)$. At first sight, when $n \gg \kappa$, our upper bound is actually better than the lower bound (6) by a $\log(n/\kappa)$ factor, which seems rather surprising and was firstly observed for a variant of SVRG [13]. [13] explained this phenomenon by the fact that SVRG does not satisfy "the span assumption" that is intrinsic for the proof in (6). However, it cannot effectively explain why SDCA [34], a variance reduction method also not satisfying the span assumption, can not have such a log-factor gain. In this paper, we provide another point of view from the sampling strategy: the randomized incremental gradient methods of [20] are referred to the ones by *sampling with replacement completely*, while the proposed VRADA algorithm is not limited to the assumption of [20]. In detail, VRADA is based on the two-loop structure of SVRG: in the outer loop, we compute the full gradient of an anchor point; in the inner loop, we compute stochastic gradients by sampling with replacement. In the outer loop of SVRG, the step of computing a full gradient can be viewed as stochastic gradient steps with $0$ step size by (implicitly) *sampling without replacement*. [8] Thus, the lower bound (6) does not apply to VRADA.

**Remark 1** (Sampling without Replacement)**.** *Very recently, the superiority of sampling without replacement has also been verified theoretically [14, 12, 27, 1]. Particularly, [1] has shown that for strongly convex and smooth finite-sum problems, SGD without replacement (also known as random reshuffling) needs $O(\sqrt{n}/\sqrt{\epsilon})$ number of stochastic gradient evaluations, which is tight and significantly better than the rate $O(1/\epsilon)$ of SGD with replacement [15]. Meanwhile, in practice, it is also more widely used in training deep neural network for its better efficiency [6, 28].*

**Other Acceleration Variants.** Besides accelerated versions of SVRG, there are a randomized primal-dual method RPDG [20], a randomized gradient extrapolation method RGEM [21], two accelerated versions Point-SAGA [9] and SSNM [9] of SAGA, and a unified approach for (random) SVRG/SAGA/SDCA/MISO [18]. In the submission of this paper, another paper [17] has also considered the approach of combining variance reduction and ADA. However, all these methods match the lower bound (6).

## 2 Algorithm: Variance Reduction via Accelerated Dual Averaging

Let $[n] := \{1, 2, \ldots, n\}$. For simplicity, we only consider the Euclidean norm $\|\cdot\| \equiv \|\cdot\|_2$. We first introduce a couple of standard assumptions about the convexity and smoothness of the problem (1).

**Assumption 1.** $\forall i \in [n]$, $g_i(\boldsymbol{x})$ *is convex,* i.e., $\forall \boldsymbol{x}, \boldsymbol{y}$, $g_i(\boldsymbol{y}) \geq g_i(\boldsymbol{x}) + \langle \nabla g_i(\boldsymbol{x}), \boldsymbol{y} - \boldsymbol{x} \rangle$; $g_i(\boldsymbol{x})$ *is* $L$*-smooth* $(L > 0)$, *i.e.,* $\|\nabla g_i(\boldsymbol{y}) - \nabla g_i(\boldsymbol{x})\| \leq L\|\boldsymbol{y} - \boldsymbol{x}\|$.

By Assumption 1 and $g(\boldsymbol{x}) = \frac{1}{n}\sum_{i=1}^{n} g_i(\boldsymbol{x})$, we can verify that $g(\boldsymbol{x})$ is $L$-smooth, *i.e.,* $\forall \boldsymbol{x}, \boldsymbol{y}, \|\nabla g(\boldsymbol{y}) - \nabla g(\boldsymbol{x})\| \leq L\|\boldsymbol{y} - \boldsymbol{x}\|$. Furthermore, we assume $l(\boldsymbol{x})$ satisfies:

**Assumption 2.** $l(\boldsymbol{x})$ *is* $\sigma$*-strongly convex* $(\sigma \geq 0)$, i.e., $\forall \boldsymbol{x}, \boldsymbol{y}$, and $l'(\boldsymbol{x}) \in \partial l(\boldsymbol{x})$, $l(\boldsymbol{y}) \geq l(\boldsymbol{x}) + \langle l'(\boldsymbol{x}), \boldsymbol{y} - \boldsymbol{x} \rangle + \frac{\sigma}{2}\|\boldsymbol{y} - \boldsymbol{x}\|^2$; *when* $\sigma = 0$, *we also say* $l(\boldsymbol{x})$ *is general convex.*

To realize acceleration with dual averaging, we recursively define the following generalized estimation sequence for the finite-sum problem (1):

$$\psi_{s,k}(\boldsymbol{z}) := \psi_{s,k-1}(\boldsymbol{z}) + a_s\big(g(\boldsymbol{y}_{s,k}) + \langle \tilde{\nabla}_{s,k}, \boldsymbol{z} - \boldsymbol{y}_{s,k} \rangle + l(\boldsymbol{z})\big), \tag{7}$$

with the initialization $\psi_{1,0}(\boldsymbol{z}) := \frac{1}{2}\|\boldsymbol{z} - \tilde{\boldsymbol{x}}_0\|^2$, $\psi_{2,0} := m\psi_{1,1}$ with $m \in \mathbb{Z}_+$, $k \in \{1, 2, \ldots, m\}$[9], and $\psi_{s+1,0} := \psi_{s,m}$ (for $s \geq 2$), where $\{a_s\}$ is a sequence of positive numbers to be specified later. Here $\{\boldsymbol{y}_{s,k}\}$ is a sequence of vectors that will be generated by our algorithm, $\{\tilde{\nabla}_{s,k}\}$ is a sequence of variance reduced stochastic gradients evaluated at $\{\boldsymbol{y}_{s,k}\}$. If $m = 1$ and $\tilde{\nabla}_{s,k} = \nabla g(\boldsymbol{y}_{s,k})$, then we can verify that (7) is equivalent to the classical definition of estimation sequence by Nesterov [26]. In the finite-sum setting, we set $m = \Theta(n)$ to amortize the computational cost per epoch $s$, where $n$ is the number of sample functions in (1). Then we say $\psi_{s,k}$ is the estimation sequence in the (inner) $k$-th iteration of the $s$-th epoch. For convenience, we define $A_s := A_{s-1} + a_s$ with $A_0 := 0$.

Based on the definition (7), Algorithm 1 summarizes the proposed *Variance Reduction via Accelerated Dual Averaging (VRADA)* method. As we see, besides the steps about updating the estimation sequence such as Steps 2-4, 6 and 12, Algorithm 1 mainly follows the framework of the simplified MiG [40] of Katyusha. The main formal differences are that we have novel and effective initialization steps in Steps 2-4 and replace the mirror descent step in MiG by Step 12, a dual averaging step:

$$\boldsymbol{z}_{s,k} := \arg\min_{\boldsymbol{z}} \psi_{s,k}(\boldsymbol{z}). \tag{8}$$

To be self-contained, note that we compute the full gradient on the anchor point in Step 7 and compute the variance reduced stochastic gradient $\tilde{\nabla}_{s,k}$ in Steps 10 and 11. Steps 9 ,12 and 14 are used to achieve acceleration. The settings for $\tilde{\boldsymbol{x}}_s$, $\boldsymbol{z}_{s+1,0}$ and $\psi_{s+1,0}$ in Step 14 are derived from our analysis. Notice that we update $\tilde{\boldsymbol{x}}_s$ as a natural uniform average with respect to $\{\boldsymbol{z}_{s,k}\}$ for both the general convex and strongly convex settings.

In Step 12, a significant characteristic is the weight $a_s$ invariant in all the inner iterations of the $s$-th epoch. As a result, at the first time, we "decouple" Nesterov's acceleration and variance reduction: the acceleration phenomenon will occur *per epoch*, while the inner iterations are mainly used for variance reduction. More precisely, the "negative momentum" in Step 9 is used to fuse the variance reduction technique into the acceleration framework, while the uniform averaging in Step 14 plays the role of "Nesterov's momentum" to achieve acceleration.

In Steps 2-4, the novel initialization steps provide us "a right way" to cancel the accumulated randomized error in the main loop: after performing a (proximal) gradient descent step, we initialize $\psi_{2,0}$ as the $m$ times of $\psi_{1,1}$. As we will see in our proof, the accumulated randomized error in the $m$ inner iterations will be completely canceled by our initialization steps.

In Step 12, due to the nature of dual averaging, we do not linearize $l(\boldsymbol{z})$ along the whole optimization trajectory. As a result, when $l(\boldsymbol{z})$ is strongly convex, it allows us to accumulate all the strong convexity in the optimization path. As we will see in the proof, this accumulation of strong convexity is crucial for our algorithm to achieve the optimal convergence rate in the regime $n \gg \kappa$.

Then we can prove (in Section 3 and Appendix) the main result for the proposed VRADA algorithm:

**Theorem 1.** *Let* $\{\tilde{\boldsymbol{x}}_s\}$ *be generated by Algorithm 1. Under Assumptions 1 and 2, and taking expectation on the randomness of all the history, we have:* $\forall s \geq 2$,

$$\mathbb{E}[f(\tilde{\boldsymbol{x}}_s)] - f(\boldsymbol{x}^*) \leq \frac{\|\tilde{\boldsymbol{x}}_0 - \boldsymbol{x}^*\|^2}{2A_s}, \tag{9}$$

**Algorithm 1** Variance Reduction via Accelerated Dual Averaging (VRADA)
___

1: **Problem:** $\min_{\boldsymbol{x}\in\mathbb{R}^d} f(\boldsymbol{x}) = g(\boldsymbol{x}) + l(\boldsymbol{x}) = \frac{1}{n}\sum_{i=1}^{n} g_i(\boldsymbol{x}) + l(\boldsymbol{x})$.
2: **Initialization:** $A_0 = 0, A_1 = a_1 = \frac{1}{L}, \boldsymbol{y}_{1,1} = \boldsymbol{z}_{1,0} = \tilde{\boldsymbol{x}}_0 \in \mathbb{R}^d, \psi_{1,0}(\boldsymbol{z}) = \frac{1}{2}\|\boldsymbol{z} - \tilde{\boldsymbol{x}}_0\|^2$.
3: $\boldsymbol{z}_{1,1} = \arg\min_{\boldsymbol{z}}\left\{\psi_{1,1}(\boldsymbol{z}) := \psi_{1,0}(\boldsymbol{z}) + a_1(g(\boldsymbol{y}_{1,1}) + \langle\nabla g(\boldsymbol{y}_{1,1}), \boldsymbol{z} - \boldsymbol{y}_{1,1}\rangle + l(\boldsymbol{z}))\right\}$.
4: $\tilde{\boldsymbol{x}}_1 = \boldsymbol{z}_{1,1}, \boldsymbol{z}_{2,0} = \boldsymbol{z}_{1,1}, \psi_{2,0} = m\psi_{1,1}$.
5: **for** $s = 2, \ldots, S$ **do**
6: $\quad A_s = A_{s-1} + \sqrt{\frac{mA_{s-1}(1+\sigma A_{s-1})}{2L}}$ and $a_s = A_s - A_{s-1}$.
7: $\quad \boldsymbol{\mu}_{s-1} = \nabla g(\tilde{\boldsymbol{x}}_{s-1})$.
8: $\quad$ **for** $k = 1, 2, \ldots, m$ **do**
9: $\quad\quad \boldsymbol{y}_{s,k} = \frac{A_{s-1}}{A_s}\tilde{\boldsymbol{x}}_{s-1} + \frac{a_s}{A_s}\boldsymbol{z}_{s,k-1}$.
10: $\quad\quad$ Sample $i$ from $\{1, 2, \ldots, n\}$ uniformly at random.
11: $\quad\quad \tilde{\nabla}_{s,k} = \nabla g_i(\boldsymbol{y}_{s,k}) - \nabla g_i(\tilde{\boldsymbol{x}}_{s-1}) + \boldsymbol{\mu}_{s-1}$.
12: $\quad\quad \boldsymbol{z}_{s,k} = \arg\min_{\boldsymbol{z}}\left\{\psi_{s,k}(\boldsymbol{z}) := \psi_{s,k-1}(\boldsymbol{z}) + a_s(g(\boldsymbol{y}_{s,k}) + \langle\tilde{\nabla}_{s,k}, \boldsymbol{z} - \boldsymbol{y}_{s,k}\rangle + l(\boldsymbol{z}))\right\}$.
13: $\quad$ **end for**
14: $\quad \tilde{\boldsymbol{x}}_s = \frac{A_{s-1}}{A_s}\tilde{\boldsymbol{x}}_{s-1} + \frac{a_s}{mA_s}\sum_{k=1}^{m}\boldsymbol{z}_{s,k}, \quad \boldsymbol{z}_{s+1,0} = \boldsymbol{z}_{s,m}, \quad \psi_{s+1,0} = \psi_{s,m}$.
15: **end for**
16: **return:** $\tilde{\boldsymbol{x}}_S$
___

*where* $\forall s \geq 2$,

$$A_s \geq \max\left\{\frac{m}{2L}\left(\frac{2}{m}\right)^{2^{-(s-1)}}, \frac{1}{L}\left(1 + \sqrt{\frac{\sigma m}{2L}}\right)^{s-1}\right\}, \tag{10}$$

*and with* $s_0 = 1 + \lceil\log_2\log_2(m/2)\rceil$, *besides the lower bounds in* (10), *we also have* $\forall s \geq s_0$,

$$A_s \geq \max\left\{\frac{m}{32L}\left(s - s_0 + 2\sqrt{2}\right)^2, \frac{m}{4L}\left(1 + \sqrt{\frac{\sigma m}{2L}}\right)^{s-s_0}\right\}. \tag{11}$$

Theorem 1 gives a unified convergence result for both the general convex ($\sigma = 0$) and strongly convex ($\sigma > 0$) settings. By (9), the objective gap is simply bounded by the term about $\|\tilde{\boldsymbol{x}}_0 - \boldsymbol{x}^*\|^2$.

To see implications of Theorem 1, by the first term in (10), whether strongly convex or not, VRADA can attain an $O\left(\frac{L}{m}\right)$-accurate solution in $1 + \lceil\log_2\log_2(m/2)\rceil$ number of epochs and thus $A_{s_0} = \frac{m}{4L}$. The superlinear phenomenon is by our novel initialization Steps 2-4 of Algorithm 1. The best known convergence rate in the initial stage is firstly obtained by SVRG$^{++}$ [4], which has shown a linear convergence rate in the initial stage for convex finite-sums. In contrast, the corresponding rate of VRADA is superlinear. To the best of our knowledge, we have not observed any theoretical justification of superlinear phenomenon for variance reduced first order methods.

By the second term in (10), in the strongly convex setting ($\sigma > 0$), we can have an accelerated linear convergence rate from the start. Note that whatever $m \leq \Theta(\kappa)$ or $m \gg \kappa$, the contracting ratio of VRADA will always be $\left(1 + \sqrt{\frac{\sigma m}{2L}}\right)^{-1}$, which will tend to 0 as $m \to +\infty$. However, for all the existing accelerated variance reduction methods such as Katyusha$^{sc}$ and Varag, when $m \gg \kappa$, the contracting ratio will be at least a constant such as $\frac{2}{3}$ in Katyusha$^{sc}$.

Then based on the prompt decrease in the superlinear initial stage, we also provide two new lower bounds for $A_s$ in (11). By the first term in (11), whether strongly convex or not, VRADA can have at least an accelerated sublinear rate. By the second term in (11), in the strongly convex setting ($\sigma > 0$), VRADA will maintain an accelerated linear rate.

Thus by Theorem 1, by setting $m = \Theta(n)$, we obtain our improved iteration complexity results for both the general convex and strongly convex settings in Table 2.

**Remark 2.** *The generalized estimation sequence* $\{\psi_{s,k}\}$ *and the associated analysis is the key in proving our main result Theorem 1, while it is commonly known to be difficult to understand. However, the estimation sequence itself also has principled explanations [11, 36]. In Section 3, we show that the estimation sequence analysis leads to a very concise, unified, and principled convergence analysis for both the general convex and strongly convex settings.*

## 3 Convergence Analysis

When using estimation sequence to prove convergence rate, the main task is to give the lower bound and upper bound of $\psi_{s,k}(z_{s,k})$, where $z_{s,k} = \arg\min_z \psi_{s,k}(z)$. The lower bound is given in terms of the objective value at the current iterate and the estimation sequence in the previous iteration, while the upper bound is in terms of the objective value at the optimal solution. (For simplicity, we only give the upper bound of $\psi_{s,m}(z_{s,m})$.) Then by telescoping and concatenating the lower bound and upper bound of $\psi_{s,k}(z_{s,k})$, we prove the rate in terms of the objective difference $f(\tilde{x}_s) - f(x^*)$ in expectation. First, in the initial Step 3 of Algorithm 1, by the smoothness property of $g(z)$ and the setting $A_1 = a_1 = \frac{1}{L}$, we have Lemma 1.

**Lemma 1** (The initial step). *It follows that* $\psi_{1,1}(z_{1,1}) \geq A_1 f(\tilde{x}_1)$.

Lemma 1 will be used to cancel the error introduced by $f(\tilde{x}_1)$ in the main loop. After entering into the main loop, by using the smoothness and convexity properties, the optimality condition of $\{z_{s,k}\}$, and the careful setting of $\{a_s\}$ and $\{A_s\}$, we obtain the lower bound of $\psi_{s,k}(z_{s,k})$ in Lemma 2.

**Lemma 2** (Lower bound). $\forall s \geq 2, k \geq 1$, *we have*

$$
\begin{aligned}
&\psi_{s,k}(z_{s,k}) \\
\geq\ & \psi_{s,k-1}(z_{s,k-1}) + a_s g(y_{s,k}) + A_s\Big(f(y_{s,k+1}) - g(y_{s,k}) - \frac{A_{s-1}}{2A_s L}\|\tilde{\nabla}_{s,k} - \nabla g(y_{s,k})\|^2\Big) \\
& -A_{s-1}\langle \tilde{\nabla}_{s,k}, \tilde{x}_{s-1} - y_{s,k}\rangle - A_{s-1}l(\tilde{x}_{s-1}).
\end{aligned}
\tag{12}
$$

In Lemma 2, the term $\|\tilde{\nabla}_{s,k} - \nabla g(y_{s,k})\|^2$ is the variance we need to bound, of which the bound is given in Lemma 3 based on the standard derivation in [2].

**Lemma 3** (Variance reduction). $\forall s \geq 2, k \geq 1$, *taking expectation on the randomness over the choice of $i$ in the $k$-th iteration of $s$-th epoch, we have*

$$
\mathbb{E}[\|\tilde{\nabla}_{s,k} - \nabla g(y_{s,k})\|^2] \leq 2L(g(\tilde{x}_{s-1}) - g(y_{s,k}) - \langle \nabla g(y_{s,k}), \tilde{x}_{s-1} - y_{s,k}\rangle).
\tag{13}
$$

Then by combining Lemmas 2 and 3, we will find that the inner product in (12) and (13) can be canceled with each other in expectation. Therefore after combining Lemmas 2 and 3, telescoping the resulting inequality from $k = 1$ to $m$ and using the definition $\psi_{s+1,0} := \psi_{s,m}$, we have Lemma 4.

**Lemma 4** (Recursion). $\forall s \geq 2$, *taking expectation on the randomness over the epoch $s$, it follows that* $\mathbb{E}[\psi_{s+1,0}(z_{s+1,0})] \geq \mathbb{E}\Big[\psi_{s,0}(z_{s,0}) + mA_s f(\tilde{x}_s) - mA_{s-1}f(\tilde{x}_{s-1})\Big]$.

Besides the lower bound in Lemma 4, by the convexity of $f(x)$ and optimality of $z_{s,m}$, we can also provide the upper bound of $\psi_{s,m}(z_{s,m})$ in Lemma 5.

**Lemma 5** (Upper bound). $\forall s \geq 2$, *taking expectation on all the history, we have*

$$
\mathbb{E}[\psi_{s,m}(z_{s,m})] \leq mA_s f(x^*) + \frac{m}{2}\|\tilde{x}_0 - x^*\|^2.
\tag{14}
$$

Finally, by combining Lemmas 1, 4 and 5, we prove Theorem 1 as follows.

**Proof of Theorem 1.**

*Proof.* Taking expectation on the randomness of all the history and telescoping Lemma 4 from 2 to $s(s \geq 2)$, we have

$$
\mathbb{E}[\psi_{s+1,0}(z_{s+1,0}) - \psi_{2,0}(z_{2,0})] \geq \mathbb{E}\Big[mA_s f(\tilde{x}_s) - mA_1 f(\tilde{x}_1)\Big],
\tag{15}
$$

where $mA_1 f(\tilde{x}_1)$ can be viewed as "accumulated randomized errors" in the main loop. It turns out that it will be cancelled by Lemma 1 and the setting $z_{2,0} = z_{1,1}, \psi_{2,0} = m\psi_{1,1}$ for our initialization steps as follows.

$$
\psi_{2,0}(z_{2,0}) = m\psi_{1,1}(z_{2,0}) = m\psi_{1,1}(z_{1,1}) \geq mA_1 f(\tilde{x}_1).
\tag{16}
$$

So combining (15) and (16), and by the setting $\psi_{s+1,0}(z_{s+1,0}) = \psi_{s,m}(z_{s,m})(s \geq 2)$, we have

$$
\mathbb{E}[\psi_{s,m}(z_{s,m})] = \mathbb{E}[\psi_{s+1,0}(z_{s+1,0})] \geq \mathbb{E}[mA_s f(\tilde{x}_s)].
\tag{17}
$$

Then combining Lemma 5 and (17), we have

$$
\mathbb{E}[mA_s f(\tilde{x}_s)] \leq \mathbb{E}[\psi_{s,m}(z_{s,m})] \leq mA_s f(x^*) + \frac{m}{2}\|\tilde{x}_0 - x^*\|^2.
\tag{18}
$$

So after simple rearrangement of (18), we obtain (9). Then by the proof of Section F, we obtain (10) and (11). $\square$

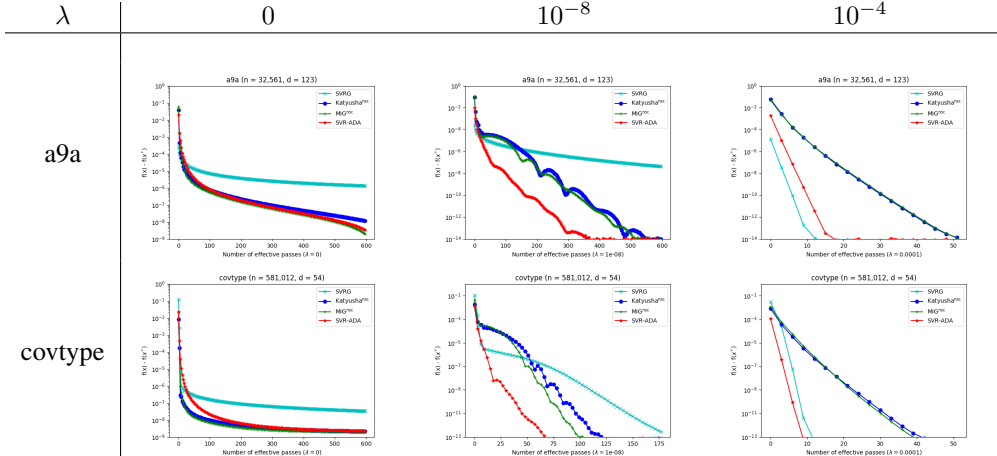

Figure 1: Comparing VRADA with SVRG, Katyusha and MiG on $\ell_2$-norm regularized logistic regression problems. The horizontal axis is the number of passes through the entire dataset, and the vertical axis is the optimality gap $f(\boldsymbol{x}) - f(\boldsymbol{x}^*)$.

## 4 Experiments

In this section, to verify the theoretical results and show the empirical performance of the proposed VRADA method, we conduct numerical experiments on large-scale datasets in machine learning. The datasets we use are a9a and covtype, downloaded from the LibSVM website[10]. To make comparison easier, we normalize the Euclidean norm of each data vector in the datasets to be $1$. The problem we study is the $\ell_2$-*norm regularized logistic regression* problem with regularization parameter $\lambda \in \{0, 10^{-8}, 10^{-4}\}$. For $\lambda = 0$, the corresponding problem is unregularized and thus general convex. For this setting, we compare VRADA with the state-of-the-art variance reduction methods SVRG [16], Katyusha$^{\text{nsc}}$ [2], and MiG$^{\text{nsc}}$ [40]. The settings $\lambda = 10^{-8}$ and $\lambda = 10^{-4}$ correspond to the strongly convex settings with a large condition number and a small one, respectively. For both settings, we compare VRADA with SVRG, Katyusha$^{\text{sc}}$ and MiG$^{\text{sc}}$.

All four algorithms we compare have a similar outer-inner structure, where we set all the number of iterations as $m = 2n$. For these algorithms, the common parameter to tune is the parameter *w.r.t.* Lipschitz constant. The details of parameter tuning can be found in Section H of the supplementary material. Our results are given in Figure 1. Following the tradition of ERM experiments, we use the number of "passes" of the entire dataset as the x-axis.

In Figure 1, when $\lambda = 0$, VRADA decreases the error promptly in the initial stage, which validates our theoretical result in attaining an $O(1/n)$-accurate solution with $\log \log n$ passes of the entire dataset. An interesting phenomenon is that the other variance reduction methods share the same behavior with VRADA in empirical evaluations (in fact, MiG$^{\text{nsc}}$ is slightly faster for both a9a and covtype datasets). This poses an open problem whether or not this superlinear phenomenon in the initial stage can be theoretically justified for SVRG, Katyusha, and MiG.

In Figure 1, when $\lambda = 10^{-8}$, *i.e.,* the large condition number setting, VRADA has significantly better performance than Katyusha$^{\text{sc}}$ and MiG$^{\text{sc}}$. This is partly due to the fact that the accumulation of strong convexity by dual averaging helps us better cancel the error from the randomness and allows VRADA to tune a more aggressive parameter about Lipschitz constant. When $\lambda = 10^{-4}$, *i.e.,* the small condition number setting, VRADA is significantly better than Katyusha$^{\text{sc}}$ and MiG$^{\text{sc}}$, which validates our superior theoretical results in (5). Meanwhile, when $\lambda = 10^{-4}$, SVRG can be competitive with VRADA, which partly verifies the theoretical results for SVRG [13].

In summary, the existing methods only perform well for the above one or two regimes, while VRADA performs well for all the three regimes: general convex, strongly convex with a large condition number and strongly convex with a small condition number, which is consistent with our theoretical results (see Table 1).

## Acknowledgements

Chaobing and Yi acknowledge support from Tsinghua-Berkeley Shenzhen Institute (TBSI) Research Fund. Yi also acknowledges support from ONR grant N00014-20-1-2002 and the joint Simons Foundation-NSF DMS grant #2031899, as well as support from Berkeley AI Research (BAIR), Berkeley FHL Vive Center for Enhanced Reality and Berkeley Center for Augmented Cognition.

## Broader Impact

The finite-sum structure widely exists in statistical learning, operational research, and signal processing. This work successfully exploits the finite-sum structure to push the performance of this kind of problems in both theory and practice. The theoretical contribution helps us better understand this simple but effective structure, while the superior empirical performance shows potential applications of this work in all the related subjects. It may benefit the broad academic and research community. There are no foreseeable negative or biased consequences.

## Footnotes

[2]For clarification, we say an algorithm is direct if it solves the problem (1) without any reformulation such as the dual reformulation [34], primal-dual reformulation [39] or warm restart reformulation [3].

[3]For instance, when the objective (1) is strongly convex, we can also use randomized coordinate descent/ascent methods on the dual or primal-dual formulation of (1) to indirectly solve (1), such as SDCA [34] and Acc-ADCA [35], APCG [24] and SPDC [39]. Variance reduction methods have also been widely applied into distributed computing [30, 22] and nonconvex optimization [29, 31].

[4]This is because, in the context of large-scale statistical learning, due to statistical limits [7, 33], even under some strong regularity conditions [7], obtaining an $O\big(\frac{1}{n}\big)$ accuracy will be sufficient.

[5]The linear convergence result is irrelevant to the problem being strongly convex or not.

[6]The rate is firstly obtained by combining Katyusha$^{sc}$ with black-box reduction, which is an indirect solver.

[7]This insightful perspective is from an anonymous NeurIPS reviewer.

[8]The outer loop of SVRG or our algorithm cannot be interpreted as stochastic gradient steps by *sampling with replacement* (say, with $0$ step size), as we cannot pick all the samples with probability $1$ by sampling with replacement for $n$ times.

[9]As we will see, $m$ denotes the number of inner iterations in our algorithm.

[10]The dataset url is https://www.csie.ntu.edu.tw/~cjlin/libsvmtools/datasets/.

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
