[Supplementary Material]

# A  Proof of Lemma 1

*Proof.* It follows that

$$
\begin{aligned}
\psi_{1,1}(\boldsymbol{z}_{1,1}) &\overset{(a)}{=} \psi_{1,0}(\boldsymbol{z}_{1,1}) + a_1(g(\boldsymbol{y}_{1,1}) + \langle \nabla g(\boldsymbol{y}_{1,1}), \boldsymbol{z}_{1,1} - \boldsymbol{y}_{1,1}\rangle + l(\boldsymbol{z}_{1,1})) \\
&\overset{(b)}{=} \frac{1}{2}\|\boldsymbol{z}_{1,1} - \boldsymbol{z}_{1,0}\|^2 + a_1(g(\boldsymbol{y}_{1,1}) + \langle \nabla g(\boldsymbol{y}_{1,1}), \boldsymbol{z}_{1,1} - \boldsymbol{y}_{1,1}\rangle + l(\boldsymbol{z}_{1,1})) \\
&\overset{(c)}{=} a_1\Big(g(\boldsymbol{y}_{1,1}) + \langle \nabla g(\boldsymbol{y}_{1,1}), \boldsymbol{z}_{1,1} - \boldsymbol{y}_{1,1}\rangle + \frac{1}{2a_1}\|\boldsymbol{z}_{1,1} - \boldsymbol{y}_{1,1}\|^2 + l(\boldsymbol{z}_{1,1})\Big) \\
&\overset{(d)}{=} a_1\Big(g(\boldsymbol{y}_{1,1}) + \langle \nabla g(\boldsymbol{y}_{1,1}), \boldsymbol{z}_{1,1} - \boldsymbol{y}_{1,1}\rangle + \frac{L}{2}\|\boldsymbol{z}_{1,1} - \boldsymbol{y}_{1,1}\|^2 + l(\boldsymbol{z}_{1,1})\Big) \\
&\overset{(e)}{\geq} a_1(g(\boldsymbol{z}_{1,1}) + l(\boldsymbol{z}_{1,1})) \\
&\overset{(f)}{=} A_1 f(\tilde{\boldsymbol{x}}_1),
\end{aligned}
$$

where $(a)$ is by definition of $\psi_{1,1}$, $(b)$ is by the definition of $\psi_{1,0}$ and $\boldsymbol{z}_{1,0} = \tilde{\boldsymbol{x}}_0$ , $(c)$ is by the setting $\boldsymbol{y}_{1,1} = \boldsymbol{z}_{1,0}$ and simple rearrangement , $(d)$ is by the setting $a_1 = \frac{1}{L}$, $(e)$ is by Lemma 6, and $(f)$ is by the setting $A_1 = a_1$ and $\tilde{\boldsymbol{x}}_1 = \boldsymbol{z}_{1,1}$. $\qquad\square$

# B  Proof of Lemma 2

*Proof.* As $l(\boldsymbol{z})$ is $\sigma$-strongly convex, by the definition of the sequence $\{\psi_{s,k}(\boldsymbol{z})\}$, $\psi_{s-1,m}(\boldsymbol{z})$ is $m + \sigma m \sum_{i=1}^{s-1} a_i = m(1 + \sigma A_{s-1})$-strongly convex. Furthermore, we also know that $\psi_{s,k}(\boldsymbol{z})(k \geq 0)$ is also at least $m(1 + \sigma A_{s-1})$-strongly convex. So it follows that: $\forall k \geq 1$,

$$
\begin{aligned}
\psi_{s,k}(\boldsymbol{z}_{s,k}) &\overset{(a)}{=} \psi_{s,k-1}(\boldsymbol{z}_{s,k}) + a_s(g(\boldsymbol{y}_{s,k}) + \langle \tilde{\nabla}_{s,k}, \boldsymbol{z}_{s,k} - \boldsymbol{y}_{s,k}\rangle + l(\boldsymbol{z}_{s,k})) \\
&\overset{(b)}{\geq} \psi_{s,k-1}(\boldsymbol{z}_{s,k-1}) + \frac{m(1 + \sigma A_{s-1})}{2}\|\boldsymbol{z}_{s,k} - \boldsymbol{z}_{s,k-1}\|^2 \\
&\quad + a_s(g(\boldsymbol{y}_{s,k}) + \langle \tilde{\nabla}_{s,k}, \boldsymbol{z}_{s,k} - \boldsymbol{y}_{s,k}\rangle + l(\boldsymbol{z}_{s,k})),
\end{aligned}
\tag{19}
$$

where $(a)$ is by the definition of $\psi_{s,k}$ and $(b)$ is by the optimality condition of $\boldsymbol{z}_{s,k-1}$ and the $m(1 + \sigma A_{s-1})$-strong convexity of $\psi_{s,k-1}$. Then we have

$$
\begin{aligned}
&a_s(g(\boldsymbol{y}_{s,k}) + \langle \tilde{\nabla}_{s,k}, \boldsymbol{z}_{s,k} - \boldsymbol{y}_{s,k}\rangle + l(\boldsymbol{z}_{s,k})) \\
&\overset{(a)}{=} a_s g(\boldsymbol{y}_{s,k}) + A_s\Big\langle \tilde{\nabla}_{s,k}, \frac{a_s}{A_s}\boldsymbol{z}_{s,k} - \boldsymbol{y}_{s,k} + \frac{A_{s-1}}{A_s}\tilde{\boldsymbol{x}}_{s-1}\Big\rangle \\
&\quad - A_{s-1}\langle \tilde{\nabla}_{s,k}, \tilde{\boldsymbol{x}}_{s-1} - \boldsymbol{y}_{s,k}\rangle + a_s l(\boldsymbol{z}_{s,k}) \\
&\overset{(b)}{\geq} a_s g(\boldsymbol{y}_{s,k}) + A_s\Big\langle \tilde{\nabla}_{s,k}, \boldsymbol{y}_{s,k+1} - \boldsymbol{y}_{s,k}\Big\rangle - A_{s-1}\langle \tilde{\nabla}_{s,k}, \tilde{\boldsymbol{x}}_{s-1} - \boldsymbol{y}_{s,k}\rangle \\
&\quad + A_s l(\boldsymbol{y}_{s,k+1}) - A_{s-1} l(\tilde{\boldsymbol{x}}_{s-1}),
\end{aligned}
\tag{20}
$$

where $(a)$ is by the fact that $A_s = A_{s-1} + a_s$ and simple rearrangement and $(b)$ is by $\boldsymbol{y}_{s,k+1} = \frac{A_{s-1}}{A_s}\tilde{\boldsymbol{x}}_{s-1} + \frac{a_s}{A_s}\boldsymbol{z}_{s,k}$ (which is by our definition of the sequence $\{\boldsymbol{y}_{s,k}\}$) and the convexity of $l(\boldsymbol{z})$.

Meanwhile, by our setting in Step 5 of Algorithm 1, $A_s = A_{s-1} + \sqrt{\frac{mA_{s-1}(1 + \sigma A_{s-1})}{2L}}$ and also $a_s = A_s - A_{s-1}$, we have

$$
\frac{mA_s(1 + \sigma A_{s-1})}{a_s^2} = \frac{2A_s}{A_{s-1}}L \geq \Big(1 + \frac{A_s}{A_{s-1}}\Big)L.
\tag{21}
$$

Then by combining ([19](#)) and ([20](#)), it follows that

$$\psi_{s,k}(\boldsymbol{z}_{s,k}) - \psi_{s,k-1}(\boldsymbol{z}_{s,k-1})$$

$$\geq \ a_s g(\boldsymbol{y}_{s,k}) + A_s\left\langle \tilde{\nabla}_{s,k}, \boldsymbol{y}_{s,k+1} - \boldsymbol{y}_{s,k}\right\rangle - A_{s-1}\langle \tilde{\nabla}_{s,k}, \tilde{\boldsymbol{x}}_{s-1} - \boldsymbol{y}_{s,k}\rangle$$

$$+ A_s l(\boldsymbol{y}_{s,k+1}) - A_{s-1} l(\tilde{\boldsymbol{x}}_{s-1}) + \frac{m(1+\sigma A_{s-1})}{2}\|\boldsymbol{z}_{s,k} - \boldsymbol{z}_{s,k-1}\|^2$$

$$\overset{(a)}{=} \ a_s g(\boldsymbol{y}_{s,k}) + A_s\left\langle \tilde{\nabla}_{s,k}, \boldsymbol{y}_{s,k+1} - \boldsymbol{y}_{s,k}\right\rangle - A_{s-1}\langle \tilde{\nabla}_{s,k}, \tilde{\boldsymbol{x}}_{s-1} - \boldsymbol{y}_{s,k}\rangle$$

$$+ A_s l(\boldsymbol{y}_{s,k+1}) - A_{s-1} l(\tilde{\boldsymbol{x}}_{s-1}) + \frac{mA_s^2(1+\sigma A_{s-1})}{2a_s^2}\|\boldsymbol{y}_{s,k+1} - \boldsymbol{y}_{s,k}\|^2$$

$$\overset{(b)}{\geq} \ a_s g(\boldsymbol{y}_{s,k}) + A_s\Bigg(\left\langle \tilde{\nabla}_{s,k}, \boldsymbol{y}_{s,k+1} - \boldsymbol{y}_{s,k}\right\rangle$$

$$+ \left(1 + \frac{A_s}{A_{s-1}}\right)\frac{L}{2}\|\boldsymbol{y}_{s,k+1} - \boldsymbol{y}_{s,k}\|^2 + l(\boldsymbol{y}_{s,k+1})\Bigg)$$

$$- A_{s-1}\langle \tilde{\nabla}_{s,k}, \tilde{\boldsymbol{x}}_{s-1} - \boldsymbol{y}_{s,k}\rangle - A_{s-1} l(\tilde{\boldsymbol{x}}_{s-1}),$$

where $(a)$ is by the fact $\boldsymbol{y}_{s,k+1} - \boldsymbol{y}_{s,k} = \frac{a_s}{A_s}(\boldsymbol{z}_{s,k} - \boldsymbol{z}_{s,k-1})$ and $(b)$ is by ([21](#)). Then we have

$$\left\langle \tilde{\nabla}_{s,k}, \boldsymbol{y}_{s,k+1} - \boldsymbol{y}_{s,k}\right\rangle + \left(1 + \frac{A_s}{A_{s-1}}\right)\frac{L}{2}\|\boldsymbol{y}_{s,k+1} - \boldsymbol{y}_{s,k}\|^2 + l(\boldsymbol{y}_{s,k+1})$$

$$= \ \left\langle \nabla g(\boldsymbol{y}_{s,k}), \boldsymbol{y}_{s,k+1} - \boldsymbol{y}_{s,k}\right\rangle + \frac{L}{2}\|\boldsymbol{y}_{s,k+1} - \boldsymbol{y}_{s,k}\|^2 + l(\boldsymbol{y}_{s,k+1})$$

$$+ \left\langle \tilde{\nabla}_{s,k} - \nabla g(\boldsymbol{y}_{s,k}), \boldsymbol{y}_{s,k+1} - \boldsymbol{y}_{s,k}\right\rangle + \frac{A_s L}{2A_{s-1}}\|\boldsymbol{y}_{s,k+1} - \boldsymbol{y}_{s,k}\|^2$$

$$\overset{(a)}{\geq} \ g(\boldsymbol{y}_{s,k+1}) - g(\boldsymbol{y}_{s,k}) + l(\boldsymbol{y}_{s,k+1}) - \frac{A_{s-1}}{2A_s L}\|\tilde{\nabla}_{s,k} - \nabla g(\boldsymbol{y}_{s,k})\|^2$$

$$= \ f(\boldsymbol{y}_{s,k+1}) - g(\boldsymbol{y}_{s,k}) - \frac{A_{s-1}}{2A_s L}\|\tilde{\nabla}_{s,k} - \nabla g(\boldsymbol{y}_{s,k})\|^2 \tag{22}$$

where $(a)$ is by Lemma [6](#) and the Young's inequality $\langle \boldsymbol{a}, \boldsymbol{b}\rangle \geq -\frac{1}{2}\|\boldsymbol{a}\|^2 - \frac{1}{2}\|\boldsymbol{b}\|^2$. So we have

$$\psi_{s,k}(\boldsymbol{z}_{s,k}) - \psi_{s,k-1}(\boldsymbol{z}_{s,k-1})$$

$$\geq \ a_s g(\boldsymbol{y}_{s,k}) + A_s\Big(f(\boldsymbol{y}_{s,k+1}) - g(\boldsymbol{y}_{s,k}) - \frac{A_{s-1}}{2A_s L}\|\tilde{\nabla}_{s,k} - \nabla g(\boldsymbol{y}_{s,k})\|^2\Big)$$

$$- A_{s-1}\langle \tilde{\nabla}_{s,k}, \tilde{\boldsymbol{x}}_{s-1} - \boldsymbol{y}_{s,k}\rangle - A_{s-1} l(\tilde{\boldsymbol{x}}_{s-1}). \tag{23}$$

$$\square$$

## C  Proof of Lemma [3](#)

*Proof.* Taking expectation on the randomness over the choice of $i$, we have

$$\mathbb{E}[\|\tilde{\nabla}_{s,k} - \nabla g(\boldsymbol{y}_{s,k})\|^2] = \ \mathbb{E}[\|\nabla g_i(\boldsymbol{y}_{s,k}) - \nabla g_i(\tilde{\boldsymbol{x}}_{s-1}) + \boldsymbol{\mu}_{s-1} - \nabla g(\boldsymbol{y}_{s,k})\|^2]$$

$$= \ \mathbb{E}[\|\nabla g_i(\boldsymbol{y}_{s,k}) - \nabla g_i(\tilde{\boldsymbol{x}}_{s-1}) + \nabla g(\tilde{\boldsymbol{x}}_{s-1}) - \nabla g(\boldsymbol{y}_{s,k})\|^2]$$

$$= \ \mathbb{E}[\|\nabla g_i(\boldsymbol{y}_{s,k}) - \nabla g_i(\tilde{\boldsymbol{x}}_{s-1})\|^2] - \|\nabla g(\tilde{\boldsymbol{x}}_{s-1}) - \nabla g(\boldsymbol{y}_{s,k})\|^2$$

$$\leq \ \mathbb{E}[\|\nabla g_i(\boldsymbol{y}_{s,k}) - \nabla g_i(\tilde{\boldsymbol{x}}_{s-1})\|^2]$$

$$\overset{(a)}{\leq} \ \mathbb{E}[2L(g_i(\tilde{\boldsymbol{x}}_{s-1}) - g_i(\boldsymbol{y}_{s,k}) - \langle \nabla g_i(\boldsymbol{y}_{s,k}), \tilde{\boldsymbol{x}}_{s-1} - \boldsymbol{y}_{s,k}\rangle)]$$

$$= \ 2L(g(\tilde{\boldsymbol{x}}_{s-1}) - g(\boldsymbol{y}_{s,k}) - \langle \nabla g(\boldsymbol{y}_{s,k}), \tilde{\boldsymbol{x}}_{s-1} - \boldsymbol{y}_{s,k}\rangle),$$

where $(a)$ is by Lemma [6](#).

$$\square$$

# D Proof of Lemma 4

*Proof.* By Lemma 2 and taking expectation on the randomness over the choice of $i$, we have

$$\mathbb{E}[\psi_{s,k}(\boldsymbol{z}_{s,k}) - \psi_{s,k-1}(\boldsymbol{z}_{s,k-1})]$$

$$\geq \quad \mathbb{E}\Big[a_s g(\boldsymbol{y}_{s,k}) + A_s\Big(f(\boldsymbol{y}_{s,k+1}) - g(\boldsymbol{y}_{s,k}) - \frac{A_{s-1}}{2A_s L}\|\tilde{\nabla}_{s,k} - \nabla g(\boldsymbol{y}_{s,k})\|^2\Big)$$

$$-A_{s-1}\langle\tilde{\nabla}_{s,k}, \tilde{\boldsymbol{x}}_{s-1} - \boldsymbol{y}_{s,k}\rangle - A_{s-1}l(\tilde{\boldsymbol{x}}_{s-1})\Big]$$

$$\overset{(a)}{\geq} \quad \mathbb{E}\Big[a_s g(\boldsymbol{y}_{s,k})$$

$$+A_s(f(\boldsymbol{y}_{s,k+1}) - g(\boldsymbol{y}_{s,k})) - A_{s-1}(g(\tilde{\boldsymbol{x}}_{s-1}) - g(\boldsymbol{y}_{s,k}) - \langle\nabla g(\boldsymbol{y}_{s,k}), \tilde{\boldsymbol{x}}_{s-1} - \boldsymbol{y}_{s,k}\rangle)$$

$$-A_{s-1}\langle\tilde{\nabla}_{s,k}, \tilde{\boldsymbol{x}}_{s-1} - \boldsymbol{y}_{s,k}\rangle - A_{s-1}l(\tilde{\boldsymbol{x}}_{s-1})\Big]$$

$$\overset{(b)}{=} \quad \mathbb{E}[A_s f(\boldsymbol{y}_{s,k+1})] - A_{s-1}f(\tilde{\boldsymbol{x}}_{s-1}), \tag{24}$$

where $(a)$ is by Lemma 3, and $(b)$ is by $\mathbb{E}[\tilde{\nabla}_{s,k}] = \nabla g(\boldsymbol{y}_{s,k})$, $A_s = A_{s-1} + a_s$ and $f(\boldsymbol{x}) = g(\boldsymbol{x}) + l(\boldsymbol{x})$.

Summing (24) from $k = 1$ to $m$, by the setting for $s \geq 2$, $\psi_{s+1,0} := \psi_{s,m}$ and $\boldsymbol{z}_{s+1,0} := \boldsymbol{z}_{s,m}$, we have

$$\mathbb{E}[\psi_{s+1,0}(\boldsymbol{z}_{s+1,0}) - \psi_{s,0}(\boldsymbol{z}_{s,0})] \quad = \quad \mathbb{E}[\psi_{s,m}(\boldsymbol{z}_{s,m}) - \psi_{s,0}(\boldsymbol{z}_{s,0})]$$

$$\geq \quad \mathbb{E}\Big[A_s \sum_{k=1}^{m} f(\boldsymbol{y}_{s,k+1}) - mA_{s-1}f(\tilde{\boldsymbol{x}}_{s-1})\Big]$$

$$\overset{(a)}{\geq} \quad \mathbb{E}\Big[mA_s f(\tilde{\boldsymbol{x}}_s) - mA_{s-1}f(\tilde{\boldsymbol{x}}_{s-1})\Big], \tag{25}$$

where $(a)$ is by the convexity of $f(\boldsymbol{z})$ and the fact of $\tilde{\boldsymbol{x}}_s = \frac{1}{m}\sum_{k=1}^{m}\boldsymbol{y}_{s,k+1}$ (which is in turn by the definition of $\tilde{\boldsymbol{x}}_s = \frac{A_{s-1}}{A_s}\tilde{\boldsymbol{x}}_{s-1} + \frac{a_s}{mA_s}\sum_{k=1}^{m}\boldsymbol{z}_{s,k}$ and the definition of $\boldsymbol{y}_{s,k}$.)

$\square$

# E Proof of Lemma 5

*Proof.* $\forall s \geq 2$, taking expectation on the choice of $i$ in the $k$-th iteration of the $s$-th epoch, we have $\forall \boldsymbol{z}$,

$$\mathbb{E}[\psi_{s,k}(\boldsymbol{z})] \quad = \quad \mathbb{E}[\psi_{s,k-1}(\boldsymbol{z}) + a_s(g(\boldsymbol{y}_{s,k}) + \langle\tilde{\nabla}_{s,k}, \boldsymbol{z} - \boldsymbol{y}_{s,k}\rangle + l(\boldsymbol{z}))]$$

$$\overset{(a)}{=} \quad \psi_{s,k-1}(\boldsymbol{z}) + a_s(g(\boldsymbol{y}_{s,k}) + \langle\nabla g(\boldsymbol{y}_{s,k}), \boldsymbol{z} - \boldsymbol{y}_{s,k}\rangle + l(\boldsymbol{z}))$$

$$\overset{(b)}{\leq} \quad \psi_{s,k-1}(\boldsymbol{z}) + a_s(g(\boldsymbol{z}) + l(\boldsymbol{z}))$$

$$= \quad \psi_{s,k-1}(\boldsymbol{z}) + a_s f(\boldsymbol{z}), \tag{26}$$

where $(a)$ is by the fact $\mathbb{E}[\tilde{\nabla}_{s,k}] = \nabla g(\boldsymbol{y}_{s,k})$, and $(b)$ is by the convexity of $g(\boldsymbol{z})$. Then taking expectation from the randomness of the epoch $s$ and telescoping (26) from $k = 1$ to $m$, we have

$$\mathbb{E}[\psi_{s,m}(\boldsymbol{z})] \quad \leq \quad \psi_{s,0}(\boldsymbol{z}) + ma_s f(\boldsymbol{z})$$

$$= \quad \begin{cases} \psi_{s-1,m}(\boldsymbol{z}) + ma_s f(\boldsymbol{z}), & s \geq 3 \\ m\psi_{1,1}(\boldsymbol{z}) + ma_2 f(\boldsymbol{z}), & s = 2. \end{cases} \tag{27}$$

Then taking expectation from the randomness of all the history from $i = 3$ and telescoping (27) to some $s \geq 3$, we have

$$\mathbb{E}[\psi_{s,m}(\boldsymbol{z})] \quad \leq \quad \psi_{2,m}(\boldsymbol{z}) + m\sum_{i=3}^{s} a_i f(\boldsymbol{z}). \tag{28}$$

Meanwhile taking expectation from the randomness of epoch $s = 2$, we have

$$
\begin{aligned}
\mathbb{E}[\psi_{2,m}(\boldsymbol{z})] &\leq m\psi_{1,1}(\boldsymbol{z}) + ma_2 f(\boldsymbol{z}) \\
&= m(\psi_{1,0}(\boldsymbol{z}) + a_1(g(\boldsymbol{y}_{1,1}) + \langle \nabla g(\boldsymbol{y}_{1,1}), \boldsymbol{z} - \boldsymbol{y}_{1,1} \rangle + l(\boldsymbol{z}))) + ma_2 f(\boldsymbol{z}) \\
&\overset{(a)}{\leq} m\Big(\frac{1}{2}\|\boldsymbol{z} - \tilde{\boldsymbol{x}}_0\|^2 + a_1(g(\boldsymbol{z}) + l(\boldsymbol{z}))\Big) + ma_2 f(\boldsymbol{z}) \\
&= m(a_1 + a_2)f(\boldsymbol{z}) + \frac{m}{2}\|\boldsymbol{z} - \tilde{\boldsymbol{x}}_0\|^2,
\end{aligned}
\tag{29}
$$

where $(a)$ is by the convexity of $g(\boldsymbol{z})$ and $\psi_{1,0}(\boldsymbol{z}) = \frac{1}{2}\|\boldsymbol{z} - \tilde{\boldsymbol{x}}_0\|^2$.

So combining (28) and (29), we have: $\forall s \geq 2$,

$$
\begin{aligned}
\mathbb{E}[\psi_{s,m}(\boldsymbol{z})] &\leq m\sum_{i=1}^{s} a_s f(\boldsymbol{z}) + \frac{m}{2}\|\boldsymbol{z} - \tilde{\boldsymbol{x}}_0\|^2 \\
&\overset{(a)}{=} mA_s f(\boldsymbol{z}) + \frac{m}{2}\|\boldsymbol{z} - \tilde{\boldsymbol{x}}_0\|^2,
\end{aligned}
\tag{30}
$$

where $(a)$ is by the our setting $a_s = A_s - A_{s-1}$ and $A_0 = 0$.

Then by (30) and the optimality of $\boldsymbol{z}_{s,m}$, we have $\psi_{s,m}(\boldsymbol{z}_{s,m}) \leq \psi_{s,m}(\boldsymbol{x}^*)$ and thus

$$
\mathbb{E}[\psi_{s,m}(\boldsymbol{z}_{s,m})] \leq \psi_{s,m}(\boldsymbol{x}^*) \leq mA_s f(\boldsymbol{x}^*) + \frac{m}{2}\|\boldsymbol{x}^* - \tilde{\boldsymbol{x}}_0\|^2.
\tag{31}
$$

$\square$

# F  The Lower Bounds for the $A_s$ in Theorem 1

*Proof.* In the following, we give the lower bound of $A_s$ by the condition in Step 6 of Algorithm 1 and $A_1 = a_1 = \frac{1}{L}$. To show the lower bound by the first term in (10), we know that

$$
A_s = A_{s-1} + \sqrt{\frac{mA_{s-1}(1 + \sigma A_{s-1})}{2L}} \geq \sqrt{\frac{mA_{s-1}(1 + \sigma A_{s-1})}{2L}} \geq \sqrt{\frac{mA_{s-1}}{2L}},
\tag{32}
$$

so we have

$$
\frac{2LA_s}{m} \geq \Big(\frac{2LA_{s-1}}{m}\Big)^{\frac{1}{2}} \geq \Big(\frac{2LA_1}{m}\Big)^{2^{-(s-1)}}.
\tag{33}
$$

Then by the setting $A_1 = \frac{1}{L}$, we have

$$
A_s \geq \frac{m}{2L}\Big(\frac{2}{m}\Big)^{2^{-(s-1)}}.
\tag{34}
$$

Meanwhile, for $s \geq 2$, we also have

$$
\begin{aligned}
A_s &\geq A_{s-1} + \sqrt{\frac{mA_{s-1}(1 + \sigma A_{s-1})}{2L}} \geq A_{s-1} + \sqrt{\frac{m\sigma}{2L}}A_{s-1} = \Big(1 + \sqrt{\frac{m\sigma}{2L}}\Big)A_{s-1} \\
&\geq \Big(1 + \sqrt{\frac{m\sigma}{2L}}\Big)^{s-1}A_1 \\
&= \frac{1}{L}\Big(1 + \sqrt{\frac{m\sigma}{2L}}\Big)^{s-1}.
\end{aligned}
\tag{35}
$$

Thus the lower bounds in (10) are proved.

Then with $s_0 = 1 + \lceil \log_2 \log_2(m/2) \rceil$, we have

$$
\begin{aligned}
A_{s_0} &\geq \frac{m}{2L}\Big(\frac{2}{m}\Big)^{2^{-(s_0-1)}} \geq \frac{m}{2L}\Big(\frac{2}{m}\Big)^{2^{-\lceil \log_2 \log_2(m/2) \rceil}} \geq \frac{m}{2L}\Big(\frac{2}{m}\Big)^{2^{-\log_2 \log_2(m/2)}} \\
&= \frac{m}{4L}.
\end{aligned}
$$

Meanwhile for $s \geq s_0 + 1$, we have

$$A_s \geq A_{s-1} + \sqrt{\frac{mA_{s-1}(1 + \sigma A_{s-1})}{2L}} \geq A_{s-1} + \sqrt{\frac{mA_{s-1}}{2L}}. \tag{36}$$

Thus we can use the mathematical induction method to prove the first lower bound in (11): $\forall s \geq s_0, A_s \geq \frac{m}{32L}\left(s - s_0 + 2\sqrt{2}\right)^2$.

Firstly, for $s = s_0$, we have $A_s \geq \frac{m}{4L} = \frac{m}{32L}(2\sqrt{2})^2$.

Then assume that for an $s \geq s_0 + 1$, $A_{s-1} \geq \frac{m}{32L}\left(s - 1 - s_0 + 2\sqrt{2}\right)^2$, then

$$
\begin{aligned}
A_s &\geq A_{s-1} + \sqrt{\frac{mA_{s-1}}{2L}} \geq \frac{m}{32L}\left(s - s_0 + 2\sqrt{2}\right)^2 + \frac{m}{16L}(s - s_0) + \frac{m}{32L}(4\sqrt{2} - 3) \\
&\geq \frac{m}{32L}\left(s - s_0 + 2\sqrt{2}\right)^2.
\end{aligned} \tag{37}
$$

Thus the first lower bound in (11) is proved.

Meanwhile, for $s \geq s_0 + 1$, we also have

$$
\begin{aligned}
A_s &\geq A_{s-1} + \sqrt{\frac{mA_{s-1}(1 + \sigma A_{s-1})}{2L}} \geq A_{s-1} + \sqrt{\frac{m\sigma}{2L}}A_{s-1} = \left(1 + \sqrt{\frac{m\sigma}{2L}}\right)A_{s-1} \\
&\geq \left(1 + \sqrt{\frac{m\sigma}{2L}}\right)^{s - s_0} A_{s_0} \\
&\geq \frac{m}{4L}\left(1 + \sqrt{\frac{m\sigma}{2L}}\right)^{s - s_0}.
\end{aligned} \tag{38}
$$

Thus the second lower bound in (11) is proved.

$\square$

# G  An Auxiliary Lemma

By Assumption 1 and [25], we have Lemma 6.

**Lemma 6.** *Under Assumption 1, $\forall \boldsymbol{x}, \boldsymbol{y}$,*

$$g(\boldsymbol{y}) \leq g(\boldsymbol{x}) + \langle \nabla g(\boldsymbol{x}), \boldsymbol{y} - \boldsymbol{x} \rangle + \frac{L}{2}\|\boldsymbol{y} - \boldsymbol{x}\|^2 \tag{39}$$

*and $\forall i \in [n], \forall \boldsymbol{x}, \boldsymbol{y}$,*

$$\|\nabla g_i(\boldsymbol{y}) - \nabla g_i(\boldsymbol{x})\|^2 \leq 2L(g_i(\boldsymbol{y}) - g_i(\boldsymbol{x}) - \langle \nabla g_i(\boldsymbol{x}), \boldsymbol{y} - \boldsymbol{x} \rangle). \tag{40}$$

Under Assumption 1, Lemma 6 are classical results in convex optimization. For completeness, we provide the proof of Lemma 6 here.

*Proof of Lemma 6.* By Assumption 1, $\forall i \in [n], g_i(\boldsymbol{x})$ satisfies $\forall \boldsymbol{x}, \boldsymbol{y}, \|\nabla g_i(\boldsymbol{x}) - \nabla g_i(\boldsymbol{y})\| \leq L\|\boldsymbol{x} - \boldsymbol{y}\|$. As a result, we have

$$
\begin{aligned}
\|\nabla g(\boldsymbol{x}) - \nabla g(\boldsymbol{y})\| &= \left\|\frac{1}{n}\sum_{i=1}^{n}\nabla g_i(\boldsymbol{x}) - \frac{1}{n}\sum_{i=1}^{n}\nabla g_i(\boldsymbol{y})\right\| \\
&\leq \frac{1}{n}\sum_{i=1}^{n}\|g_i(\boldsymbol{x}) - g_i(\boldsymbol{y})\| \\
&\leq L\|\boldsymbol{x} - \boldsymbol{y}\|.
\end{aligned} \tag{41}
$$

The we have

$$
\begin{aligned}
g(\boldsymbol{y}) &= g(\boldsymbol{x}) + \int_0^1 \langle \nabla g(\boldsymbol{x} + \tau(\boldsymbol{y} - \boldsymbol{x})), \boldsymbol{y} - \boldsymbol{x} \rangle d\tau \\
&= g(\boldsymbol{x}) + \langle \nabla g(\boldsymbol{x}), \boldsymbol{y} - \boldsymbol{x} \rangle + \int_0^1 \langle \nabla g(\boldsymbol{x} + \tau(\boldsymbol{y} - \boldsymbol{x})) - \nabla g(\boldsymbol{x}), \boldsymbol{y} - \boldsymbol{x} \rangle d\tau. \quad (42)
\end{aligned}
$$

Then it follow that

$$
\begin{aligned}
g(\boldsymbol{y}) - g(\boldsymbol{x}) - \langle \nabla g(\boldsymbol{x}), \boldsymbol{y} - \boldsymbol{x} \rangle &\leq \left| \int_0^1 \langle \nabla g(\boldsymbol{x} + \tau(\boldsymbol{y} - \boldsymbol{x})) - \nabla g(\boldsymbol{x}), \boldsymbol{y} - \boldsymbol{x} \rangle d\tau \right| \\
&\leq \int_0^1 |\langle \nabla g(\boldsymbol{x} + \tau(\boldsymbol{y} - \boldsymbol{x})) - \nabla g(\boldsymbol{x}), \boldsymbol{y} - \boldsymbol{x} \rangle| \, d\tau \\
&\leq \int_0^1 \|\nabla g(\boldsymbol{x} + \tau(\boldsymbol{y} - \boldsymbol{x})) - \nabla g(\boldsymbol{x})\| \|\boldsymbol{y} - \boldsymbol{x}\| d\tau \\
&\leq \int_0^1 L\tau \|\boldsymbol{y} - \boldsymbol{x}\|^2 d\tau \\
&= \frac{L}{2} \|\boldsymbol{y} - \boldsymbol{x}\|^2. \quad (43)
\end{aligned}
$$

Thus we obtain (39).

Then denote $\forall i \in [n], \phi_i(\boldsymbol{y}) = g_i(\boldsymbol{y}) - g_i(\boldsymbol{x}) - \langle \nabla g_i(\boldsymbol{x}), \boldsymbol{y} - \boldsymbol{x} \rangle$. Obviously $\phi_i(\boldsymbol{y})$ is also $L$-smooth. One can check that $\nabla g_i(\boldsymbol{x}) = 0$ and so that $\min_{\boldsymbol{y}} \phi_i(\boldsymbol{y}) = \phi_i(\boldsymbol{x}) = 0$, which implies that

$$
\begin{aligned}
\phi_i(\boldsymbol{x}) &\leq \phi_i\left(\boldsymbol{y} - \frac{1}{L}\nabla\phi_i(\boldsymbol{y})\right) \\
&= \phi_i(\boldsymbol{y}) + \int_0^1 \left\langle \nabla\phi_i\left(\boldsymbol{y} - \frac{\tau}{L}\nabla\phi_i(\boldsymbol{y})\right), -\frac{1}{L}\nabla\phi_i(\boldsymbol{y}) \right\rangle d\tau \\
&= \phi_i(\boldsymbol{y}) + \left\langle \nabla\phi_i(\boldsymbol{y}), -\frac{1}{L}\nabla\phi_i(\boldsymbol{y}) \right\rangle + \int_0^1 \left\langle \nabla\phi_i\left(\boldsymbol{y} - \frac{\tau}{L}\nabla\phi_i(\boldsymbol{y})\right) - \nabla\phi_i(\boldsymbol{y}), -\frac{1}{L}\nabla\phi_i(\boldsymbol{y}) \right\rangle d\tau \\
&\leq \phi_i(\boldsymbol{y}) - \frac{1}{L}\|\nabla\phi_i(\boldsymbol{y})\|^2 + \int_0^1 L \left\| \frac{\tau}{L}\nabla\phi_i(\boldsymbol{y}) \right\| \left\| \frac{1}{L}\nabla\phi(\boldsymbol{y}) \right\| d\tau \\
&\leq \phi_i(\boldsymbol{y}) - \frac{1}{2L}\|\nabla\phi_i(\boldsymbol{y})\|^2. \quad (44)
\end{aligned}
$$

Then we have $\|\nabla\phi_i(\boldsymbol{y})\|^2 \leq 2L(\phi_i(\boldsymbol{y}) - \phi_i(\boldsymbol{x}))$. Then by the definition of $\phi_i(\boldsymbol{y})$, we obtain (40). $\quad\square$

## H  Experimental Details and Supplementary Experiments

Besides running binary classification experiments on the two datasets a9a and covtype, we also run multi-class classification experiments on mnist and cifar10. The problem we solve is the $\ell_2$-*norm regularized (multinomial) logistic regression* problem:

$$
\min_{\boldsymbol{w}\in\mathbb{R}^{d\times(c-1)}} f(\boldsymbol{w}) := \frac{1}{n}\sum_{j=1}^n \left( -\sum_{i=1}^{c-1} y_j^{(i)} \boldsymbol{w}^{(i)^T}\boldsymbol{x}_j + \log\left(1 + \sum_{i=1}^{c-1}\exp\left(\boldsymbol{w}^{(i)^T}\boldsymbol{x}_j\right)\right)\right) + \frac{\lambda}{2}\sum_{i=1}^{c-1}\|\boldsymbol{w}^{(i)}\|_2^2,
$$

$$(45)$$

where $n$ is the number of samples, $c \in \{2, 3, \ldots\}$ denotes the number of class (for a9a and covtype, $c = 2$; for mnist and cifar10, $c = 10$.), $\lambda \geq 0$ denotes the regularization parameter, $\boldsymbol{y}_j = (y_j^{(1)}, y_j^{(2)}, \ldots, y_j^{(c-1)})^T$ is a one-hot vector or zero vector[11], and $\boldsymbol{w} := (\boldsymbol{w}^{(1)}, \boldsymbol{w}^{(2)}, \ldots, \boldsymbol{w}^{(c-1)}) \in \mathbb{R}^{d\times(c-1)}$ denotes the variable to optimize. For the two-class datasets "a9a" and "covtype", we have presented our results by choosing the regularization parameter $\lambda \in \{0, 10^{-8}, 10^{-4}\}$. For the ten-class datasets "mnist" and "cifar10", we choose $\lambda \in \{0, 10^{-6}, 10^{-3}\}$.

Figure 2: Comparing VRADA with SVRG, Katyusha and MiG on $\ell_2$-norm regularized multinomial logistic regression problems. The horizontal axis is the number of passes through the entire dataset, and the vertical axis is the optimality gap $f(\boldsymbol{x}) - f(\boldsymbol{x}^*)$.

For the four algorithms we compare, the common parameter to tune is the parameter *w.r.t.* Lipschitz constant[12], which is tuned in $\{0.0125, 0.025, 0.05, 0.1, 0.25, 0.5\}$.[13] All four algorithms are implemented in C++ under the same framework, while the figures are produced using Python.

As we see, despite there are some minor differences among different tasks/datasets shown in Figure 1 and Figure 2, the general behaviors are still very consistent. From both figures, our method VRADA is competitive with other two accelerated methods, and is much faster than the non-accelerated SVRG algorithm in the general convex setting and the strongly convex setting with a large conditional number. Meanwhile, in the strongly convex setting with a small conditional number, VRADA is still competitive with the non-accelerated SVRG algorithm and much faster than the other two accelerated algorithms of Katyusha[sc] and MiG[sc].

## Footnotes

[11]Zero vector denotes the class of the $j$-th sample is $c$.

[12]For logistic regression with normalized data, the Lipschitz constant is globally upper bounded [39] by $1/4$, but in practice we can use a smaller one than $1/4$.

[13]In our experiments, due to the normalization of datasets, all the four algorithms will diverge when the parameter is less than $0.0125$. Otherwise, they always converge if the parameter is less than $0.5$.