[Reviews · NeurIPS 2020]

Review 1

Summary and Contributions: The authors present a method that combines several ideas, namely acceleration + dual averaging + stochastic variance reduction, to solve the "finite sum" problem with regularization. The authors present a theory for the proposed approach that improves upon some known methods in the literature (see Table 1). Numerical results are presented on a couple of data sets from LIBSVM.

Strengths: + By combining several ideas into a framework, the authors are able to recover (and improve upon) several known methods in the literature in terms of complexity results. + The problem studied is an important problem in data analysis.

Weaknesses: - The paper is not clearly written in many places. I give some examples below under "clarity". - Is the idea of putting together different ideas (dual averaging + variance reduction + acceleration) truly novel? Of course, very few papers are truly novel, but here I am getting at whether there are any new ideas learned. I do not see anything "new" arise from the paper. - The numerical results are not very convincing. The proposed method (SVR-ADA) performs about the same as two competing methods in the first column of Figure 1, it performs better in the second column, and it performs worse in the third column. Not to mention, only two data sets were chosen to be displayed. Granted, the authors do provide two other datasets in the supplementary material, but there (Figure 2) SVR-ADA performs worse relative to Figure 1. Specifically, SVR-ADA performs roughly the same in column 2, worse in column 1, and marginally better (maybe) in column 1. - Why is only two-norm regularization used in the numerical results, as opposed to one-norm, for example? Overall, the range of problems tested is rather small. - Why lambda in {0, 10^-8, 10^-4} in Figure 1, but then lambda in {0, 10^-6,10^-3} in Figure 2 of the supplementary material?

Correctness: The main claims and theoretical results in the paper are likely to be correct to the best of my knowledge, although I did not read the proofs.

Clarity: The paper is not clearly written in many places. Here are a few examples. - The abstract is difficult to understand. It is not clear that the claims are true, although perhaps it is just because it is not clearly written. Does "SVR-ADA improve the complexity of the best known methods ..."? Some of the best known methods have provably optimal worst-case complexity, so that seems impossible. It appears that what you mean is that some of the methods in Table 1 are improved upon. - "Nonstrongly convex" in line 20 just appears to be "convex". Why not use that? Moreover, you assume that l in line 19 is sigma-strongly convex for sigma >= 0, but then in line 20 when sigma = 0 you call it nonstrongly convex. - Line 30: this is not true since \nabla g_i(x) is not even a random variable, so taking the expectation makes no sense. It is, of course, true that one can give a correct definition of what a stochastic gradient is, and for the finite sum problem a stochastic gradient can be obtained by choosing, at random, a single element of the sum. That is very different from what is written in line 30. - Line 32: again, f(x) is not a random variable, so the expectation makes no sense. You probably mean E(f(x_k) - f(x_*), where x_k (the kth iteration of the random process) is a random variable. To be clear, I am not implying that the authors do not know these things, but rather that the authors did not clearly write various parts of the paper that will end up confusing the reader, especially those that may not already be an expert in the area.

Relation to Prior Work: The relationship of this work to prior work on variance reduced methods is clearly described. However, a bit of a broader context with respect to other of the most well known methods that do not use variance reduction could be given a bit more attention.

Reproducibility: Yes

Additional Feedback: I have read the response by the authors. I have increased my "overall score" because the author response did convince me more that there is enough merit in the improved theoretical results. The comments concerning the numerical results did not change my original assessment of that aspect of the paper.


Review 2

Summary and Contributions: UDPATE AFTER THE AUTHOR RESPONSE: ================================= I have read the author response as well as the other reviews, and I would like to keep my higher score. I explain the reason below. Before going over the reasons, though, I would like to point out two issues: 1. The derivations to get (9) definitely have to be included, and it also has to be made clear that the second and third bound in (9) hold only for $s \ge s0$. The current form of (9) doesn't hold, e.g., for $s \eq 2$. 2. Unlike what is claimed in the author response, the authors are indeed using Nesterov's momentum, not just negative momentum. The difference is that Nesterov's momentum typically uses the most recent iterate as a guess of the next iterate (hence, e.g., the way [26] performs accelerated DA), whereas in this work, the authors are using the iterate $z_{s,k}$ (Line 9) as a guess of the next *epoch-level* iterate, which is indeed the uniform average of these initial guesses. In both cases, the gradient is taken at the average of the previous iterates together with the guess. This difference, I think, is worth being pointed out. The analysis of accelerated SVRG is far from a trivial, incremental combination of the previous work. Here is a list of challenges that has to be overcome: 1- Actually how should you "add" momentum to SVRG? -------------------------------------------------- Specifically, there are two iterates in SVRG: $x_t$ and $\tl{x}_s$. There are two other iterates in acceleration: the output $x_t$ of the base algorithm (more on that below), and the extrapolated point $y_t$ which is a weighted average of the previous $x_t$ (taking the gradient estimate at $y_t$). How do you combine SVRG and acceleration then? Would you average $x_t$, $\tl{x}_t$, or both, and by exactly what weights? This paper, to my knowledge for the first time, analyzes what seems to be the *right* way to do this: "simply change the weights at the epoch-level rather than inside the epochs." And then, this results in a simple, uniform averaging of the iterates at the end of the epoch. This is also intuitive: the job of the inner loop is to reduce variance, but acceleration should happen mainly at the outer-loop (Lines 9 and 14 of the Algorithm). The answers of all previous work to this is counter-intuitive. For example, [R1] averaged all the iterates $x_t$ as well as a negative momentum to come up with $y_t$, and then had to use a complicated, non-uniform weighting of the iterates at the end of each epoch to form $\tl{x}_t$. Similar complications are there in other works. See, for example, the rather complicated weighting that VARAG [Lan et al. 2019, cited in the review] has to use to get things to work. 2- What should be the length of the epochs? ------------------------------------------- This is also quite a non-trivial choice when you want to get optimal bounds. VARAG and [R1] (the only direct methods obtaining the optimal bounds up to log(n) terms) have to use a completely non-trivial epoch-length schedule which is half SVRG++ and half SVRG (i.e., exponentially increasing, until a given threshold). This paper uses the simple and intuitive fixed epoch length $m$ (like basic SVRG), without any difficulty arising in the proof. 3- Should we use Mirror-Descent (MD) or Dual-Averaging (DA) as the base? ------------------------------------------------------------------------ Katyusha seems to combine both updates, VARAG chooses MD, while [R1] uses DA, though in a limited setting. Yet, the analyses in all the above papers is much more complicated than what is provided in the current submission. --- The above challenges are effectively overcome in this work, and the solutions would be of great value to those working on such methods, even if the paper had only provided bounds similar to the previous work. That it actually achieves a theoretical improvement despite simpler algorithm and analysis, and still remains empirically competitive at least in some small examples, is quite remarkable. The simplicity of the algorithm is a great advantage when you want to implement it. The algorithm is almost as simple as SVRG and Nesterov's algorithm alone. AFAIK, we simply didn't have such an accelerated SVRG algorithm before. Refs: - [R1] Joulani et al., ICML-2020 (see "Relation to prior work" below). =================================== The paper provides a new accelerated variance-reduced algorithm for optimizing finite-sums. The algorithm is based on SVRG and Dual-Averaging, with Nesterov-style acceleration. The main technique is to limit the Nesterov extrapolation to the outer loop of the SVRG epochs, while the final output of each epoch is a simple uniform average of the iterates within that epoch, and the query point is a much simpler, straightforward "negative-momentum" combination (compared to previous work). This makes the algorithm remarkably simple. The authors then provide a very simple, unified proof that captures both the strongly-convex and non-strongly-convex settings. As a result, the algorithm obtains a rate for non-strongly-convex functions that is almost optimal, and better than all the previous work I am aware of; there are also improvements for the strongly-convex setting.

Strengths: The paper provides an almost-optimal algorithm for a problem for which previous algorithms have been: a) much harder to analyze; b) much harder to implement; c) suffering from worse convergence bounds; d) limited in applicability / optimality to either non-strongly-convex or strongly-convex objectives, not both. This paper resolves all those issues. The proof and algorithm are remarkably simple. Thus, the paper's contributions are indeed significant, both for theory and for practice.

Weaknesses: The major weakness is the fact that one previous paper, including a very directly relevant result, is missed. In particular, there should be a complete comparison, both in terms of the convergence regimes and in terms of the techniques, with the work of Lan et al. (NeurIPS-2019) before any publication of the current paper; see below. Other issues: - The fact that this apparently applies only to the unconstrained setting: It would be great if you could add a discussion of what has to change to apply this algorithm and convergence bound in the constrained setting.

Correctness: I have checked the proofs in detail, and they do seem completely correct. The proofs are also very well-written and succinct, despite the fact that the details of most steps are provided. Having said that, the derivations to obtain (9), and then (10) and (11) should also be provided for completeness. Here are a few typos to fix in the proofs: - Line 375: "Step 5" -> "Step 6" - (26): $\nabla$ missing on the second line (Also: perhaps you should mention that this is a standard result in the SVRG literature, and cite the relevant works.) - (27): Expectation sign missing from the r.h.s. of the first inequality There are several more typos and grammatical issues throughout the paper, please consider a second proof-reading.

Clarity: The paper is very well-written.

Relation to Prior Work: There is one recent work that the authors may not be aware of, which is very relevant to the results in this paper. In particular: - Lan et al. (NeurIPS-2019). "A unified variancereduced accelerated gradient method for convex optimization" In this paper, the authors provide a unified algorithm and analysis which is similar, but more complicated, than the one provided in the present paper. Unlike this work, their algorithm is based on Mirror-Descent rather than Dual-Averaging, and they obtain a bound that depends on $n log n$ rather than $n log(log(n))$. A complete comparison with their convergence regimes is definitely missing and should be added to this paper. ( On that note, there is also a recent ICML paper which you may find relevant - though, of course, it came out after this NeurIPS submission: - Joulani et al. (ICML-2020). "A simpler approach to accelerated optimization: iterative averaging meets optimism" This uses a regret-based analysis and online learning algorithms to get a convergence bound for the non-strongly-convex setting of order $n log(n) + \sqrt{nL/\epsilon}$. A difference is that for this algorithm the guarantee holds not only for the output of an epoch, but also for the running iterate at any point in time. This is also based on dual-averaging. )

Reproducibility: Yes

Additional Feedback: In the author response, please provide a detailed comparison with the work of Lan et al. (2019) in terms of both the convergence regimes and the techniques.


Review 3

Summary and Contributions: This paper introduces stochastic variance reduced accelerated dual averaging (SVR-ADA) for finite-sum convex optimization. The paper provides the convergence analysis for both convex and strongly-convex settings. The paper provides numerical validation of SVR-ADA on toy regularized logistic regression problems. The experiments are quite limited, and I doubt the real advantage and applicability of SVR-ADA in large-scale machine learning.

Strengths: This paper is well-written, and the related work is well-presented. Theoretical guarantees for the proposed SVR-ADA have been studied in both convex and strongly-convex scenarios, which is decent. Some numerical experiments are carried out to verify the theoretical results, and from this viewpoint, this paper is complete.

Weaknesses: 1. SVR-ADA does not outperform the baseline methods; in particular, for CIFAR10 (see supplementary). It is completely unacceptable to report a few selected results on simple applications. From this viewpoint, there is no advantage of the proposed SVR-ADA over Katyusha. 2. This paper builds on top of the well-known control-variate approach to reduce the variance of the stochastic gradient and proposed the SVR-ADA algorithm. The control variate approach has been widely studied in many different scenarios, but the application is quite limited, which in practice cannot outperform the baseline SGD. The novelty of this work is very limited, and the contribution is very incremental. 3. More discussion is required to explain the theoretical advantage of SVR-ADA over the existing work. I did not see the real advantage of the theoretical results listed in Table 1 compared with the existing variance reduction methods. 4. In modern machine learning, most of the objective functions are nonconvex, and deep neural networks are used. Whether SVR-ADA applicable to training deep neural networks should be studied, and the generalization performance should be addressed.

Correctness: The claims and methods seem correct to me.

Clarity: The paper is well-written and easy to follow.

Relation to Prior Work: The paper does a good job in the literature review.

Reproducibility: Yes

Additional Feedback: Please see the weaknesses section. ========================== Post rebuttal I have read the author's feedback. The authors ignored most of my important comments; in particular the experiments. I believe this paper need a lot of more effort in order to be published at the flagship venue of the machine learning conference.


Review 4

Summary and Contributions: This paper gives a new optimization algorithm SVR-ADA for finite sum convex optimization. It improves the existing convergence rates of previous works in certain settings. For example, for non-strongly convex and smooth functions, it improves has a convergence rate of nloglog(n) compared to nlog(n) which was the previous best when epsilon=1/n. They also give improvements for the strongly convex case when n >> kappa.

Strengths: This paper gives a new optimization algorithm SVR-ADA for finite sum convex optimization. It improves the existing convergence rates of previous works in certain settings. For example, for non-strongly convex and smooth functions, it improves has a convergence rate of nloglog(n) compared to nlog(n) which was the previous best when epsilon=1/n. They also give improvements for the strongly convex case when n >> kappa also shows that accelerated methods can be strictly better than non-accelerated methods. The paper also mentions that their algorithm is simple and has a unified analysis for both the convex and strongly convex settings. They also give experimental evaluation for their method.

Weaknesses: I think the regimes in which the algorithm improves on existing bounds is fairly restricted. Moreover, the paper improves log(n) to loglog(n) which might not be a very significant improvement since log(n) is also typically small in practice. Minor Comments and typos: 1) The legend and axis labels are very small and hard to read in the paper. 2) line 201: given -> give 3) line 90: full -> fully 4) line 390: \tilde{x}_{s+1} - > \tilde{x}_{s} 5) In equation 27, second line, parenthesis is misplaced. 6) line 381 third equation -> g(x_{s-1}) -> nabla g(x_{s-1}) 7) On line 363, equation 1, why do we have \nabla g(y_1,1) instead of the variance reduced gradient? ------------ Based on the discussion with other reviewers and the authors' response, I have increased my score because the paper presents a simple to analyze novel method of combining SVRG and acceleration which is theoretically interesting.

Correctness: The claims seem correct.

Clarity: See comments above.

Relation to Prior Work: Yes.

Reproducibility: Yes

Additional Feedback:

[Author Response · NeurIPS 2020]

To **Reviewers**, we will make all suggested minor corrections in the final version and address main concerns below.

**R1** (and **R3, R5:**) Thanks for your constructive comments! The idea of integrating several key optimization techniques,
dual averaging + variance reduction + acceleration, is novel and nontrivial for three reasons. **First**, for the general convex
setting, the proposed SVR-ADA gives the rate $O\big(n \log \log n + \frac{\sqrt{nL}}{\sqrt{\epsilon}}\big)$, which improves the SOTA rate $O\big(n \log \frac{1}{\epsilon} + \frac{\sqrt{nL}}{\sqrt{\epsilon}}\big)$
of Katyusha, not substantially improved for over $4$ years; for the well-conditioned strongly convex setting (the case
of the number of samples $n$ far greater the condition number, *i.e.*, $n \gg \kappa$), SVR-ADA has the rate $O\big(n \log \log n + $
$\frac{n}{\log(n/\kappa)} \log \frac{1}{n\epsilon}\big)$ which improves the SOTA rate $O\big(n \log \frac{1}{\epsilon}\big)$ of SAG, unchanged for 8 years; for the ill-conditioned
strongly convex setting ($n \leq O(\kappa)$), SVR-ADA matches the lower bound in [25]. **Second**, besides the improved or
optimal convergence results, SVR-ADA shares the simplicity of MiG [29] and the unification of Varag (see comments
of **R2**) *simultaneously*. **Third**, this work is the first to show that in the finite sum setting, combining accelerated dual
averaging (ADA) with variance reduction (VR) gives better convergence rates than that of combining accelerated mirror
descent (AMD) with VR as adopted in Katyusha, MiG and Varag. This provides new perspectives to acceleration.

In terms of experiments, SVR-ADA is compared with SOTA finite sum solvers. The results send one consistent
message: *SVR-ADA performs well in all the three settings, namely general convex, ill-conditioned or well-conditioned*
*strongly convex*. In comparison, the existing accelerated algorithms Katyusha and MiG do not perform so well for
the well-conditioned strongly convex setting; the non-accelerated algorithm SVRG does not perform well for the
general convex and ill-conditioned strongly convex settings. In terms of the choice of regularization, we use different
two-norm regularizations to represent the phenomenon of the three settings compactly by Figure 1 and Figure 2.
If we use one-norm, then it can only represent the general convex setting. For completeness, we will show the
numerical results of one-norm in the final version. The different regularization parameters of the two figures are used to
better represent the obvious change in experimental phenomena. In the final version, we will represent the results of
$\{0, 10^{-8}, 10^{-7}, 10^{-6}, 10^{-5}, 10^{-4}, 10^{-3}\}$ for both experiments at least in the supplementary material.

Thanks for your suggestions about writing! In the final version, we will rewrite the abstract to make it more clear. Per
your suggestions, we will use "general convex" to replace "nonstrongly convex" and make the statements in Line 30 and
32 more precise. We will add a paragraph to make a simple introduction to stochastic algorithms that do not use VR.

**R2:** We are very glad that you obviously recognize significance of our contributions! We were also aware of Varag
after our submission! In the final version, we will make systematical comparisons with Varag from three perspectives:
1) **Efficiency.** SVR-ADA improves the SOTA rates for general convex and well-conditioned strongly convex settings,
while Varag does not improve any SOTA rates. 2) **Simplicity.** SVR-ADA only uses two-point coupling in the inner
iteration, while Varag uses three-point coupling. SVR-ADA uses a uniform average in the outer iteration, while Varag
uses a weighted one. 3) **Unification.** SVR-ADA unifies the general convex and strongly convex settings with much
simplified parameter settings, which can be simply specified in algorithm description. However, to adapt to both settings,
the parameter settings of Varag are very complicated and not explicitly and clearly stated in the algorithmic description.

In terms of techniques, **first**, we use the combination of VR and ADA, while they use the combination of VR and AMD;
**second**, we use completely different and concise convergence analysis by estimation sequence; **third**, we only use
negative momentum, while they use both Nesterov's momentum and negative momentum.

In terms of applicability in the constrained setting, we can always reformulate a convex constrained convex problem as
an unconstrained convex problem by the indicator function of the constrained set. Thus if the indicator function admits
an efficient proximal operator, then we can apply SVR-ADA in the constrained setting. Meanwhile, in the final version,
we will give a complete proof for the bound of $A_k$. We will cite the latest reference Joulani et al. (ICML 2020)!

**R3:** Thanks for your constructive comments! We fully understand your concern with the applicability for nonconvex
problems such as deep neural networks. However, it is an open problem for all control variate finite sum solvers, not
only for ours. Meanwhile, the *primary area* of our contribution is in convex optimization, which is valuable for many
problems in machine learning, signal processing, operational research. Thus we cannot undervalue the widely studied
control variate approaches as they significantly reduce the overall complexity for finite sum convex optimization. It is
fair to say broad applicability is the main reason why the control variate finite sum solvers are so widely studied!

In terms of **theoretical contributions**, as highlighted in our response to **R1**, we have improved rates in two regimes
that remain unimproved for many years, in a very actively studied area. In terms of **empirical performance**, as in our
response to **R1**, SVR-ADA is the first algorithm that performs well in all the three settings. We will conduct more
experimental evaluation as per the request by you and **R1** and report in the final version.

**R5:** Thanks for your constructive comments! For the general convex setting, the best-known result $O(n \log \frac{1}{\epsilon} + \frac{\sqrt{nL}}{\sqrt{\epsilon}})$ is
optimal up to a $\log$ factor. In convex optimization, a main endeavor is to shave off $\log$ factors to match the corresponding
lower bounds. We have made a substantial step forward by reducing $\log n$ to $\log \log n$. Meanwhile, as in our response
to **R1**, we also improve the rate for the well-conditioned strongly convex setting. As **R2** commented, the simplicity and
unification merits of SVR-ADA are remarkable! We believe our work is of significant value to this area.

[Meta-Review · NeurIPS 2020]

The paper presents a novel way to combine SVRG and acceleration for finite-sum convex optimization: an algorithm based on dual averaging is presented together with a unified analysis which achieves an almost optimal rate for non-strongly convex and smooth objectives, and also provides improvements for strongly convex objectives. Previous algorithms for this problem are much harder to analyze and implement, suffer from worse convergence bounds, and are limited in applicability/optimality to either non-strongly-convex or strongly-convex objectives, not both. This makes the results of the paper significant. At the same time, the authors are encouraged to extend their experiments and revise their interpretation (e.g., take back on the claim that the new algorithm outperforms the baselines, as it is not the case in all the experiments, as pointed out by Reviewer 3).